# Hebbian plasticity in parallel synaptic pathways: A circuit mechanism for systems memory consolidation

**Michiel W. H. Remme**[1☯¤a], **Urs Bergmann**[1☯¤b], **Denis Alevi**[2,3],
**Susanne Schreiber**[1,3,4], **Henning Sprekeler**[2,3,4,5‡], **Richard Kempter**[1,3,4‡]*

**1** Department of Biology, Institute for Theoretical Biology, Humboldt-Universität zu Berlin, Berlin, Germany,
**2** Department for Electrical Engineering and Computer Science, Technische Universität Berlin, Berlin,
Germany, **3** Bernstein Center for Computational Neuroscience Berlin, Berlin, Germany, **4** Einstein Center for
Neurosciences Berlin, Berlin, Germany, **5** Excellence Cluster *Science of Intelligence*, Berlin, Germany

☯ These authors contributed equally to this work.
¤a Current address: INAIT, Lausanne, Switzerland
¤b Current address: Google Research, Berlin, Germany
‡ HS and RK also contributed equally to this work.
* r.kempter@biologie.hu-berlin.de

pcbi.1009681

UNITED KINGDOM

**Data Availability Statement:** All relevant data are
within the paper. The relevant code to generate the

## Abstract

Systems memory consolidation involves the transfer of memories across brain regions and
the transformation of memory content. For example, declarative memories that transiently
depend on the hippocampal formation are transformed into long-term memory traces in neo-
cortical networks, and procedural memories are transformed within cortico-striatal networks.
These consolidation processes are thought to rely on replay and repetition of recently
acquired memories, but the cellular and network mechanisms that mediate the changes of
memories are poorly understood. Here, we suggest that systems memory consolidation
could arise from Hebbian plasticity in networks with parallel synaptic pathways—two ubiqui-
tous features of neural circuits in the brain. We explore this hypothesis in the context of hip-
pocampus-dependent memories. Using computational models and mathematical analyses,
we illustrate how memories are transferred across circuits and discuss why their representa-
tions could change. The analyses suggest that Hebbian plasticity mediates consolidation by
transferring a linear approximation of a previously acquired memory into a parallel pathway.
Our modelling results are further in quantitative agreement with lesion studies in rodents.
Moreover, a hierarchical iteration of the mechanism yields power-law forgetting—as
observed in psychophysical studies in humans. The predicted circuit mechanism thus brid-
ges spatial scales from single cells to cortical areas and time scales from milliseconds to
years.

## Author summary

After new memories are acquired, they can be transferred over time into other brain
areas—a process called systems memory consolidation. For example, new declarative

results of this paper can be found at https://github.com/sprekelerlab/Remme-Bergmann-2021.

**Funding:** This work was funded by the German Research Foundation (DFG, https://www.dfg.de/, project number 327654276 - SFB 1315 to HS, SS, and RK), the German Federal Ministry of Education and Research (BMBF, https://www.bmbf.de, Bernstein Award FKZ GQ1201 to HS; 01GQ1705 to RK; 01GQ0901 and 01GQ1403 to SS), and the Einstein Foundation Berlin (https://www.einsteinfoundation.de, to MR and SS). The funders had no role in study design, data collection and analysis, decision to publish, or preparation of the manuscript.

memories, which refer to the conscious memory of facts and events, depend on the hippocampus. Older declarative memories, however, also rely on neocortical networks. The cellular mechanisms underlying such a transfer are poorly understood. In this work, we show that a simple and in the brain ubiquitous connectivity pattern, combined with a standard learning rule, leads to gradual memory transfer. We illustrate our proposed mechanism in numerical simulations and mathematical analyses. At the neurophysiological level, our theory explains experimental findings on memory storage in the hippocampal formation when specific pathways between neural populations are disrupted. At the psychophysical level, we can account for the power-law forgetting curves typically found in humans. A consequence of the proposed model is that consolidated memories can yield faster responses because they are stored in increasingly shorter synaptic pathways between sensory and motor areas. By giving a mechanistic explanation of the consolidation process, we contribute to the understanding of the transfer of memories and the reorganization of memories over time.

## Introduction

Clinical and lesion studies suggest that declarative memories initially depend on the hippocampus, but are later transferred to other brain areas [1–3]. Some forms of memory eventually become independent of the hippocampus and depend only on a stable representation in the neocortex [1–3]. Similarly, procedural memories are consolidated within cortico-striatal networks [1, 4, 5]. This process of memory transformation—termed systems memory consolidation—is thought to prevent newly acquired memories from overwriting old ones, thereby extending memory retention times ("plasticity-stability dilemma"; [6–10]), and to enable a simultaneous acquisition of episodic memories and semantic knowledge of the world [11, 12]. While specific neuronal activity patterns, including for example an accelerated replay of recent experiences [13, 14], are involved in the transfer of memories from hippocampus to neocortex [15], the mechanisms underlying systems memory consolidation are not well understood. Specifically, it is unclear how this consolidation-related transfer is shaped by the anatomical structure and the plasticity of the underlying neural circuits. This poses a substantial obstacle for understanding into which regions memories are consolidated; why some memories are consolidated more rapidly than others [16–18]; why some memories stay hippocampus dependent, and why and how the character of memories changes over time [1]; and whether the consolidation of declarative and non-declarative memories [1, 4, 5] are two sides of the same coin. These questions are hard to approach within phenomenological theories of systems consolidation such as the standard consolidation theory [11, 19], the multiple trace theory [16], and the trace transformation theory [20, 21]. Here, we propose a novel mechanistic foundation of the consolidation process that accounts for several experimental observations and that could contribute to understanding the transfer of memories and the reorganisation of memories over time on a neuronal level.

Our focus lies on simple forms of memory that can be phrased as cue-response associations. We assume that such associations are stored in synaptic pathways between an input area—neurally representing the cue—and an output area—neurally representing the response. Thus, our work relates to feedforward, hetero-associative memory (and is therefore applicable to both declarative and non-declarative memories) rather than recurrent, auto-associative memory (see, e.g., [22–24]). Our central hypothesis—the parallel pathway theory (PPT)—is that systems memory consolidation arises naturally from the interplay of two abundantly found

neuronal features: parallel synaptic pathways and Hebbian plasticity [25, 26]. First, we illustrate this theory in a simple hippocampal circuit motif and show that Hebbian plasticity can consolidate previously stored associations into parallel pathways. Next, we outline the PPT in a mathematical framework for the simplest possible (linear) case. Then we show in simulations that the proposed mechanism is robust to various neuronal nonlinearities; further, the mechanism reproduces the results of a hippocampal lesion study in rodents [27]; iterated in a cascade, it can achieve a full consolidation into neocortex and result in power-law forgetting of memories as is observed in psychophysical studies in humans [28].

## Results

### A mechanistic basis for systems memory consolidation

The suggested parallel pathway theory (PPT) relies on a parallel structure of feedforward connections onto the same output area: a direct, monosynaptic and an indirect, multisynaptic pathway. We propose that memories are initially stored in the indirect pathway and are subsequently transferred to the direct pathway via Hebbian plasticity. Because the indirect pathway is multisynaptic, it transmits signals with a longer time delay than the direct pathway (Fig 1A). A timing-dependent plasticity rule allows the indirect pathway to act as a teacher for the direct pathway.

The proposed mechanism can be exemplified in the hippocampal formation, by considering direct and indirect pathways to area CA1. CA1 receives a direct, monosynaptic pathway from the entorhinal cortex (EC), which is called perforant path ($PP_{CA1}$, Fig 1B, red; [29]). In addition, EC input is relayed to CA1 via the classical trisynaptic pathway via dentate gyrus (DG) and CA3, reaching CA1 through the Schaffer collaterals (SC; Fig 1B, blue; [29]).

As in earlier theories, we assume that the indirect pathway via CA3 is involved in the original storage of memories [30, 31], an assumption that is supported by experiments, e.g. [32–34]. We neglect, for simplicity, any encoding-related change in the direct pathway, even though in animals this pathway might also show some, putative much lower, plasticity during memory acquisition. This simplification does not affect our proposed mechanism on the consolidation-related transfer of memories.

We assume encoding in such a way that a memory can be recalled by a specific neural activity pattern in EC—a cue—that triggers spikes in a subset of CA1 cells through this indirect pathway via the SC, representing the associated response. The same cue reaches CA1 also through the direct pathway via the $PP_{CA1}$. We assume that this direct input from EC initially fails to trigger spikes because the synaptic weight pattern in the $PP_{CA1}$ does not match the cue. However, $PP_{CA1}$ inputs that are activated by the cue precede the spikes in CA1 pyramidal cells that are triggered by the indirect pathway by 5–15 ms [35] due to transmission delays. Presynaptic spikes preceding postsynaptic spikes with a short delay favor selective long-term potentiation by spike timing-dependent plasticity (STDP, Fig 1C) [36–38]. Consequently, cue-driven $PP_{CA1}$ synapses onto activated CA1 cells are strengthened until the memory that was initially stored in the indirect pathway can be recalled via the direct pathway alone. The indirect pathway thus acts as a teacher for the direct pathway.

To illustrate this mechanism, we used a simple integrate-and-fire neuron model (for details, see Methods) of a CA1 cell that receives inputs through the SC and the $PP_{CA1}$. We also considered the two pathways to contain the same number of synapses and transmit identical spike patterns apart from a 5-ms delay in the SC (Fig 1D). Consolidation then corresponds to copying the synaptic weight pattern of the SC to the $PP_{CA1}$. In line with our hypothesis, such a consolidation was indeed achieved by STDP in the $PP_{CA1}$ synapses (Fig 1E). A consolidation in the opposite direction, i.e., from the $PP_{CA1}$ to the SC cannot be achieved by STDP because the

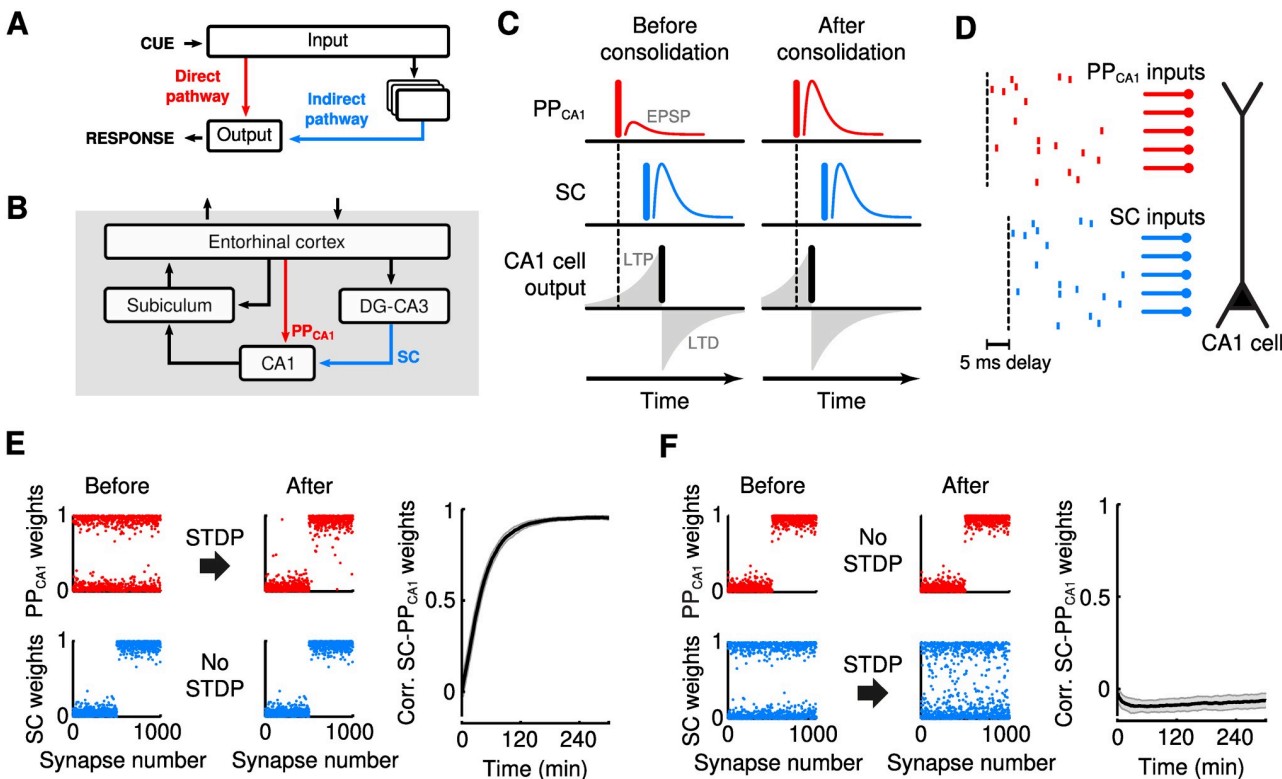

**Fig 1. A mechanistic basis for systems memory consolidation.** (A) Circuit motif for the parallel pathway theory. Cue-response associations are initially stored in an indirect synaptic pathway (blue) and consolidated into a parallel direct pathway (red). (B) Hippocampal connectivity. The entorhinal cortex projects to CA1 through an indirect pathway via DG-CA3 and the Schaffer collaterals (SC, blue arrow), and through the direct perforant path ($PP_{CA1}$, red arrow). (C) Model of consolidation through STDP. Left: before consolidation, a strong SC input (middle, blue vertical bar) causes a large EPSP and triggers a spike in CA1 (bottom, black vertical bar). A weak $PP_{CA1}$ input (top, red) that precedes the SC input is potentiated by STDP. Right: after consolidation through STDP, the $PP_{CA1}$ input (top) can trigger a spike in CA1 by itself (bottom). (D-E) Consolidation in a single integrate-and-fire CA1 cell receiving 1000 $PP_{CA1}$ and 1000 SC excitatory inputs. (D) $PP_{CA1}$ activity consists of independent poisson spike trains; the SC activity is an exact copy of the $PP_{CA1}$ activity, delayed by 5 ms. (E) Consolidation of a synaptic weight pattern from non-plastic SC synapses to plastic $PP_{CA1}$ synapses. Left and middle: normalized synaptic weights before and after consolidation. Right: time course of correlation between SC and $PP_{CA1}$ weight vectors during consolidation (mean ± SEM for 10 trials). (F) Failure of consolidation of a synaptic weight pattern from non-plastic $PP_{CA1}$ to plastic SC synapses; panels as in E.

temporal order of spiking activity is reversed and hence does not favour synaptic potentiation (Fig 1F). Note that in this simple example, the EC-to-DG/CA3 synapses don't store any memory, but only introduce the transmission delay. In the following, we will show that all synapses of the indirect pathway can be involved in the original storage of memories.

To understand the conditions under which the suggested PPT can achieve a consolidation of associative memories, we performed a mathematical analysis, which shows that consolidation is robust to differences in the neural representation in the two pathways and illustrates its dependence on the temporal input statistics in the two pathways. Readers who are less interested in the mathematical details are welcome to jump to section "Consolidation of spatial representations", where we show in simulations that the mechanism is robust to neuronal complexities; in subsequent sections, we also show that the mechanism accounts for lesion studies in rodents, and that it can be hierarchically iterated.

**Theory of spike timing-dependent plasticity (STDP) for parallel input pathways.** In the following mathematical analysis, we consider a single cell that receives inputs through two pathways, as in Fig 1A. The cell could be located, for example, in CA1, as in Fig 1B. We assume

that memories, i.e., cue-response associations, are stored in the synaptic weight vector $\mathbf{V}$ of the indirect path, and that consolidation occurs by transferring this information into the weights $\mathbf{W}$ of the direct path. In the simulation in Fig 1, the weight vector $\mathbf{V}$ represents the SC pathway, and the vector $\mathbf{W}$ the $PP_{CA1}$ pathway. For simplicity, we consider the case of a single rate-based neuron, which represents one of the output neurons in the simulated network. Very similar theoretical results can be obtained for the spiking case of linear Poisson neurons, apart from additional contributions from spike-spike correlations, which can be neglected for a large number of synapses [39].

The output $y$ of the rate-based neuron is assumed to be given by a linear function of the input

$$y(t) = \mathbf{W}^\mathsf{T}\mathbf{x}(t) + \mathbf{V}^\mathsf{T}\mathbf{x}'(t - D) \tag{1}$$

where the vectors $\mathbf{x}$ and $\mathbf{x}'$ denote the input arising from the direct and indirect pathways, respectively, and $\mathsf{T}$ denotes the transpose of a vector (or matrix). We assume that the inputs $\mathbf{x}$ and $\mathbf{x}'$ are both representations of the cue and therefore are related by some kind of (potentially nonlinear) statistical dependency. Moreover, we assume that $\mathbf{x}'$ arises from an indirect pathway and is therefore delayed by a time interval $D > 0$. The notation is chosen such that the case where the two inputs to the two pathways are the same (apart from the delay) reduces to the condition $\mathbf{x}(t) = \mathbf{x}'(t)$, which is the case, e.g., in Fig 1D.

We now consider the learning dynamics of a simple additive (STDP) rule that would result from a rate picture (neglecting spike-spike correlations; cf. [39]),

$$\frac{\Delta\mathbf{W}}{T} = \eta \int_{-\infty}^{\infty} \mathrm{d}\tau \frac{1}{T} \int_0^T \mathrm{d}t\, L(\tau)\, \mathbf{x}(t)\, y(t + \tau) = \eta \int_{-\infty}^{\infty} \mathrm{d}\tau\, L(\tau)\, \langle \mathbf{x}(t)y(t+\tau)\rangle_T \ , \tag{2}$$

where $L(\tau)$ is the learning window (example in section Effects of temporal input statistics on systems memory consolidation), which determines how much a pair of pre- and postsynaptic activity pulses (i.e., spikes) with a time difference $\tau$ changes the synaptic weight, and $\eta$ is a learning rate that scales the size of these changes. We adopt the convention that the time difference $\tau$ is positive when a presynaptic spike occurs before a postsynaptic spike.

The notation $\langle \cdots \rangle_T = \frac{1}{T}\int_0^T \cdots \mathrm{d}t$ indicates averaging over an interval of length $T$. We assume that the integration time $T$ can be chosen such that the weights do not change significantly during the integration time (i.e., a small learning rate $\eta$), but that the statistics of the input are sufficiently well sampled so that boundary effects in the temporal integration are negligible. We also assume that the statistics of the inputs $\mathbf{x}$ and $\mathbf{x}'$ are stationary, i.e., they do not change over time. Under these assumptions, we can insert the output firing rate $y$ from Eq (1) into the learning rule in Eq (2) and get

$$\begin{aligned}
\frac{\Delta\mathbf{W}}{T} &= \eta \int_{-\infty}^{\infty} \mathrm{d}\tau\, L(\tau)\langle \mathbf{x}(t)[\mathbf{W}^\mathsf{T}\mathbf{x}(t+\tau) + \mathbf{V}^\mathsf{T}\mathbf{x}'(t+\tau-D)]\rangle_T \\
&\approx \eta\left[\int_{-\infty}^{\infty} \mathrm{d}\tau\, L(\tau)\, \langle \mathbf{x}(t)\mathbf{x}(t+\tau)^\mathsf{T}\rangle_t\right]\mathbf{W} \\
&\quad + \eta\left[\int_{-\infty}^{\infty} \mathrm{d}\tau\, L(\tau)\, \langle \mathbf{x}(t)\mathbf{x}'(t+\tau-D)^\mathsf{T}\rangle_t\right]\mathbf{V}
\end{aligned} \tag{3}$$

where $\langle \cdots \rangle_t$ denotes the average over *all* times. Eq (3) describes the dynamics of the weights $\mathbf{W}$ in the direct pathway, which are driven by an interplay of the correlation structures within the direct pathway (through $\langle \mathbf{x}(t)\mathbf{x}(t+\tau)^\mathsf{T}\rangle_t$) and between the two pathways (through

$\langle \mathbf{x}(t)\mathbf{x}'(t + \tau - D)^{\mathsf{T}}\rangle_t$); the dynamics of $\mathbf{W}$ depends also on the shape of the learning window $L$ and the weights $\mathbf{V}$ in the indirect pathway.

It is important to emphasize that in this analysis of the learning dynamics we consider the input arising during consolidation, e.g., during sleep, and this input may be statistically different from the input during memory storage or recall. If the correlation structure between the two pathways is different during consolidation and during storage/recall, the consolidation process leads to a distortion of the memory in the sense that a different cue would be required to retrieve the memory. Here, we consider only the case where the correlation structure during consolidation is the same as during storage and recall.

Let us now study under which conditions this weight update generates a consolidation of the input-output associations stored initially in the weights $\mathbf{V}$ of the indirect pathway into the weights $\mathbf{W}$ of the direct pathway.

**Learning dynamics implement memory consolidation as a linear regression.** In general, the learning dynamics is hard to analyze if the covariance matrices $\langle \mathbf{x}(t)\mathbf{x}(t + \tau)^{\mathsf{T}}\rangle_t$ and $\langle \mathbf{x}(t)\mathbf{x}'(t + \tau - D)^{\mathsf{T}}\rangle_t$ are arbitrary objects. A case that can be studied analytically is that of separable statistics in which each of the two correlation matrices can be written as a product of scalar functions $f$ and $g$ of the delay $\tau$ and the covariance matrices for zero delay, $\langle \mathbf{x}\mathbf{x}^{\mathsf{T}}\rangle$ and $\langle \mathbf{x}\mathbf{x}'^{\mathsf{T}}\rangle$:

$$\langle \mathbf{x}(t)\mathbf{x}(t + \tau)^{\mathsf{T}}\rangle_t =: \langle \mathbf{x}\mathbf{x}^{\mathsf{T}}\rangle f(\tau) \tag{4}$$

$$\langle \mathbf{x}(t)\mathbf{x}'(t + \tau - D)^{\mathsf{T}}\rangle_t =: \langle \mathbf{x}\mathbf{x}'^{\mathsf{T}}\rangle g(\tau - D) \tag{5}$$

For simplicity, we omitted the lower index $t$ in $\langle \mathbf{x}\mathbf{x}^{\mathsf{T}}\rangle$ and $\langle \mathbf{x}\mathbf{x}'^{\mathsf{T}}\rangle$. Note that this separability assumption is consistent with all simulations shown, except the one in the section "Consolidation of spatial representations" of the Results; see there for details.

For separable input statistics, the learning dynamics in Eq (3) can be simplified to

$$\frac{1}{\eta}\frac{\Delta \mathbf{W}}{T} = \underbrace{\left[\int_{-\infty}^{\infty} \mathrm{d}\tau\, L(\tau)f(\tau)\right]}_{=:A} \langle \mathbf{x}\mathbf{x}^{\mathsf{T}}\rangle \mathbf{W} + \underbrace{\left[\int_{-\infty}^{\infty} \mathrm{d}\tau\, L(\tau)g(\tau - D)\right]}_{=:B} \langle \mathbf{x}\mathbf{x}'^{\mathsf{T}}\rangle \mathbf{V}. \tag{6}$$

If the scalar constant $A$ is negative (see below for conditions when this is the case), the learning dynamics is stable and converges to a unique fixed point that is given by

$$\mathbf{W} = -\frac{B}{A}[\langle \mathbf{x}\mathbf{x}^{\mathsf{T}}\rangle]^{-1}\langle \mathbf{x}\mathbf{x}'^{\mathsf{T}}\rangle \mathbf{V}. \tag{7}$$

Note that apart from the factor $-\frac{B}{A} =: \beta$, this fixed point has the same structure as the closed-form solution of a linear regression. In fact, it is straightforward to show that the learning dynamics in Eq (6) performs a gradient descent on the error function

$$E(\mathbf{W}) := \langle (\mathbf{W}^{\mathsf{T}}\mathbf{x} - \beta\, \mathbf{V}^{\mathsf{T}}\mathbf{x}')^2\rangle_t. \tag{8}$$

If $A$ is negative and $B$ is positive (and thus $\beta$ is positive), the learning dynamics in the direct path converges to a weight configuration for which the input $\mathbf{W}^{\mathsf{T}}\mathbf{x}$ from the direct path is an optimal linear approximation of the input $\mathbf{V}^{\mathsf{T}}\mathbf{x}'$ from the indirect path, in the sense of minimal mean squared error $E$. If $\beta > 1$, the direct pathway would contribute more to a potential recall than the original memory trace in the indirect pathway. A sign reversal of $\beta$ (i.e. $\beta < 0$) implies a sign reversal of $\mathbf{V}$. Then, however, a constraint on the sign change of weights (see below for details) would prohibit consolidation; memories in the indirect path could even be actively

deleted from the direct path. In summary, memory consolidation in the PPT is supported by $A < 0$ (stable dynamics) and $B > 0$ (consolidation possible), which implies $\beta := -\frac{B}{A} > 0$.

Note that we assumed storage of original memories only in the weight vector $V$ representing the SC pathway. But since learning in the direct pathway is driven by the input from the entire indirect pathway, these results also hold if original memories are stored in any other plastic synapses of the indirect pathway (e.g. EC to DG/CA3 in Fig 1B).

Let us relate the theoretical results obtained so far to the simulations shown in Fig 1; although the simulations are performed for integrate-and-fire neurons, our theory on rate-based neurons accounts for the main findings: Because two inputs $\mathbf{x}$ and $\mathbf{x}'$ are the same apart from the delay, the fixed point condition Eq (7) reduces to $\mathbf{W} = -\frac{B}{A}\mathbf{V}$, in line with the result that the weights are copied into the direct pathway. Because the learning window is dominated by depression we have $A < 0$ while the delay in combination with the shorter autocorrelation time of the Poisson processes in the input ensures $B > 0$. A consolidation from the direct to the indirect pathway is not possible because this inverts the delay and pushes the cross-correlation between the two pathways into the depression component of STDP. As a result, the factor $B$ is negative and consolidation fails.

In terms of systems memory consolidation in general, the weights $\mathbf{V}$ of the indirect path change as new memories are acquired, so the fixed point in Eq (7) for the weights $\mathbf{W}$ of the direct path is usually never reached. If it were, the direct pathway would merely represent a copy of the memories that are currently stored in the indirect path rather than retaining older memories, as intended. The time scale of the learning dynamics of the direct path [determined by $\eta$ in Eq (6)] should therefore be longer than the memory retention time in the indirect path, which is determined, e.g., by the rate at which new memories are stored. In case of a small enough $\eta$, the transient dynamics of the system is more important for the consolidation process than the fixed point.

Another important aspect to emphasize is that the consolidation is influenced by the correlation structure $\langle \mathbf{x}\mathbf{x}'^{\mathsf{T}} \rangle$ between the two pathways that is encountered during the consolidation period. Intuitively and according to Eq (8), consolidation is achieved by matching the input $\mathbf{V}^{\mathsf{T}}\mathbf{x}'$ that is caused by "cues" $\mathbf{x}'$ in the indirect path with the input $\mathbf{W}^{\mathsf{T}}\mathbf{x}$ caused by the associated "cues" $\mathbf{x}$ in the direct path. In order for the consolidated memories to be accessible during recall, the relation between the "cues" in the two pathways (i.e., the correlation $\langle \mathbf{x}\mathbf{x}'^{\mathsf{T}} \rangle$ between the two pathways) should be the same during recall as during consolidation.

The objective function argument in Eq (8) only holds when the constant $A$ is negative. For positive $A$, the learning dynamics in Eq (7) suffers from the common Hebbian instability and thus has to be complemented by a weight-limiting mechanism. The choice of this weight limitation (e.g., subtractive or divisive normalization, weight bounds) will then have an impact on the dynamics and the fixed point of the learning process [40, 41]. For the simulations, the parameters were therefore always chosen such that the learning dynamics were stable ($A < 0$). Although this suggests that no weight limiting mechanism was required in principle, upper and lower bounds for the weights were nevertheless used in simulations, with no qualitative impact on the results.

**Effects of temporal input statistics on systems memory consolidation.** The constants $A$ and $B$, which were defined in Eq (6) as $A := \int d\tau\, L(\tau) f(\tau)$ and $B := \int d\tau\, L(\tau) g(\tau - D)$, play an important role for the learning dynamics. As already elaborated, the sign of $A$ determines stability while $B$ should be positive to obtain consolidation. Sign and magnitude depend on the interplay between the learning window $L$ and the temporal input statistics, characterized by the correlation functions $f$ and $g$ defined in Eqs (4) and (5), respectively. For the assumed

separable statistics, $f$ is fully determined by the autocorrelation of the input in the direct path, and $f(\tau)$ is therefore symmetric in time $\tau$.

A first interesting observation is that for the special case of an antisymmetric learning window $L$, we obtain $A = 0$ for symmetry reasons. Mathematically, this implies that the first term of the learning dynamics in Eq (6)—the dependence of the change of the weights **W** in the direct path on their actual value—vanishes. Intuitively, the balance of potentiation and depression in an antisymmetric learning window implies that the direct path, although able to drive the postsynaptic neuron, is causing equal amounts of potentiation and depression in all of its synapses. On average, synaptic changes are caused only by the indirect pathway with weights **V**, which therefore acts as a supervisor for the learning dynamics of **W** in the direct path. A thorough analysis under which conditions STDP can be used for supervised learning has been provided elsewhere [42, 43], and the results of this analysis are applicable in the present case. Functionally, the depressing part of an STDP learning window serves to neutralize the impact of the direct pathway on its own learning dynamics, effectively creating a supervised learning scenario.

Another interesting observation relates to the magnitude of the terms $A$ and $B$, which is determined by the time scale on which the inputs change (reflected, e.g., in the time constants of the decay of the correlation functions $f$ and $g$). Let us assume that both correlation functions $f(\tau)$ and $g(\tau)$ are maximal for $\tau = 0$ and that they decay to 0 for large $|\tau|$; such conditions are reasonable for most correlation structures. We also assume that the learning window has the typical structure of potentiation for causal timing, $L(\tau) > 0$ for $\tau > 0$, and depression for acausal timing, $L(\tau) < 0$ for $\tau < 0$ [36, 37, 44]. Then the delay $D > 0$ in the indirect path shifts the maximum of the cross-correlation $g(\tau - D)$ into the potentiating part of the learning window (Fig 2B) while the maximum of $f(\tau)$ remains in the transition region of potentiation and depression (Fig 2A). The following three observations can be made concerning the constant $B$ as defined by the integral in Eq (6):

(1). If the cross-correlation $g$ has a narrow enough peak (i.e., narrower than the time scale of the learning window and the delay $D$), $B$ is positive, suggesting that consolidation can occur (Fig 2B). The sharp localization of $g$ corresponds to rapidly changing input signals.

(2). If the decay time constant of the cross-correlation $g$ is large compared to that of the learning window, the depressing component of the learning window has more impact and reduces the constant $B$ and thus the efficiency of consolidation (Fig 2C). In the case where the learning window is dominated by depression, $B$ can even get negative for large time constants of $g$, abolishing consolidation altogether.

(3). If the delay $D$ along the indirect path is much longer than the decay time constant of the learning window, we obtain $B \approx 0$, meaning that consolidation is abolished (Fig 2D). In other words, the delayed correlations between the two pathways are too large to be exploited by STDP. This will limit the ability to consolidate from too long indirect paths into shortcuts.

## Consolidation of spatial representations

The mathematical analysis of the PPT makes two key predictions. First, it suggests that STDP in a parallel direct pathway achieves consolidation by performing a linear regression between inputs in the direct and the indirect pathways [Eqs (7) and (8)]. Therefore, the proposed mechanism should generalize to situations in which the cue representations in the direct and indirect pathways differ. Second, the theory suggests that consolidation is most

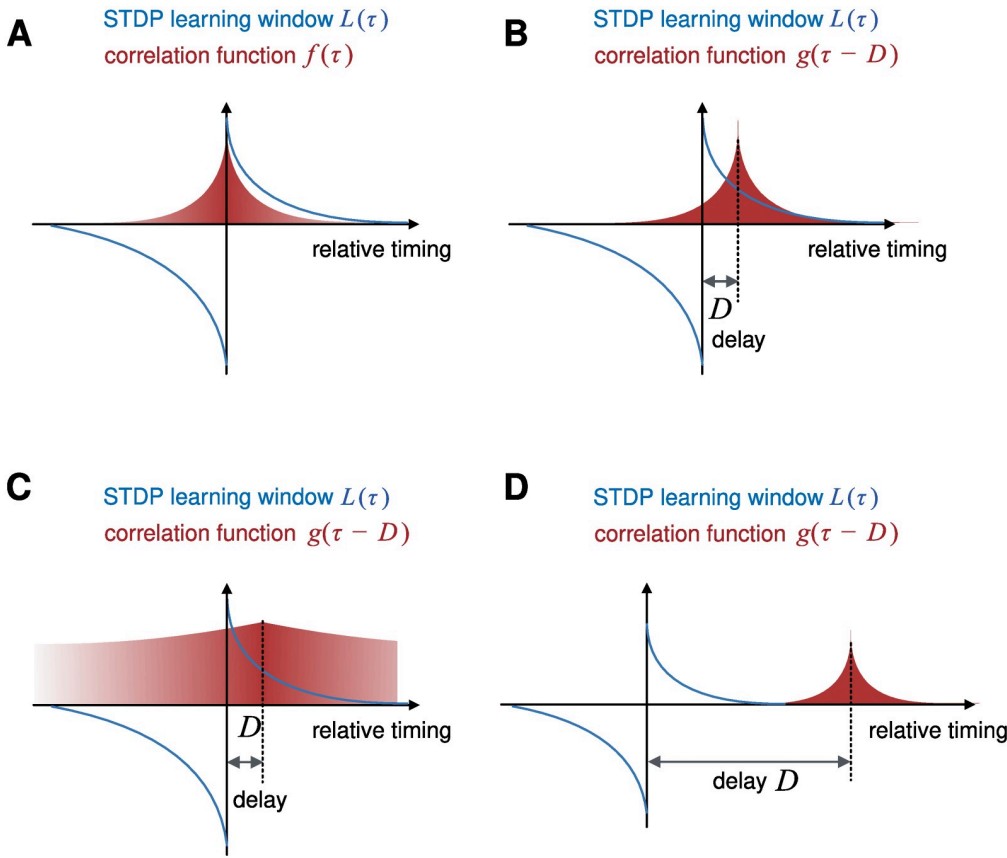

**Fig 2. Interaction of temporal correlations and the STDP learning window.** The weight dynamics of the direct path [Eq (6)] is driven by inputs from the direct and indirect paths: weight changes are determined by the integrated products of the STDP learning window $L$ with the autocorrelation $f$ [Eq (4)] and the cross-correlation $g$ [Eq (5)], respectively. (A) Examples of a learning window $L(\tau)$ and an autocorrelation $f(\tau)$, both plotted as a function of the "relative timing" $\tau$. For separable statistics, $f$ is symmetric. If the learning window $L$ has a stronger negative part for $\tau < 0$ and a weaker positive part for $\tau > 0$, the coefficient $A := \int d\tau\, L(\tau)f(\tau)$ is typically negative. (B)–(D) Learning window $L$ as in (A) and three example cross-correlations $g$. (B) The indirect path primarily induces potentiation in the direct path if $B := \int d\tau\, L(\tau)g(\tau - D) > 0$. This is the case if (i) the delay $D$ between the paths is positive, (ii) the learning window is positive for positive delays, and (iii) the time scale of the decay of cross-correlations $g$ is shorter than the delay $D$ and the width of the learning window $L$. These three conditions favor consolidation. (C) If the cross-correlation $g$ decays on a time scale that is much longer than the width of the learning window and the delay $D$, the indirect path can drive both potentiation and depression, and consolidation is weaker (i.e., the coefficient $B$ is smaller) than for shorter correlations. (D) If the delay $D$ between the direct and the indirect paths is longer than the width of the learning window $L$, the indirect path cannot induce systematic changes in the weights of the direct path (coefficient $B \approx 0$), and consolidation is ineffective.

effective when the correlation time constants of the input during consolidation is matched to the coincidence time scale of STDP (Fig 2B). In the following, we will show in simulations that those predictions hold and, moreover, that the mechanism is robust to neuronal nonlinearities.

To begin with, we show that the mechanism is robust to differing cue representations in the two pathways and to weaker correlations among them [45]. To this end, we used place cell representations [46] for the SC input from CA3 and grid cell representations [47, 48] for the $PP_{CA1}$ input from EC (Fig 3A). Moreover, we show that the suggested mechanism is compatible with the biophysical properties of CA1 neurons, which receive inputs in different subcellular compartments. To this end, we simulated a multicompartmental CA1 pyramidal cell

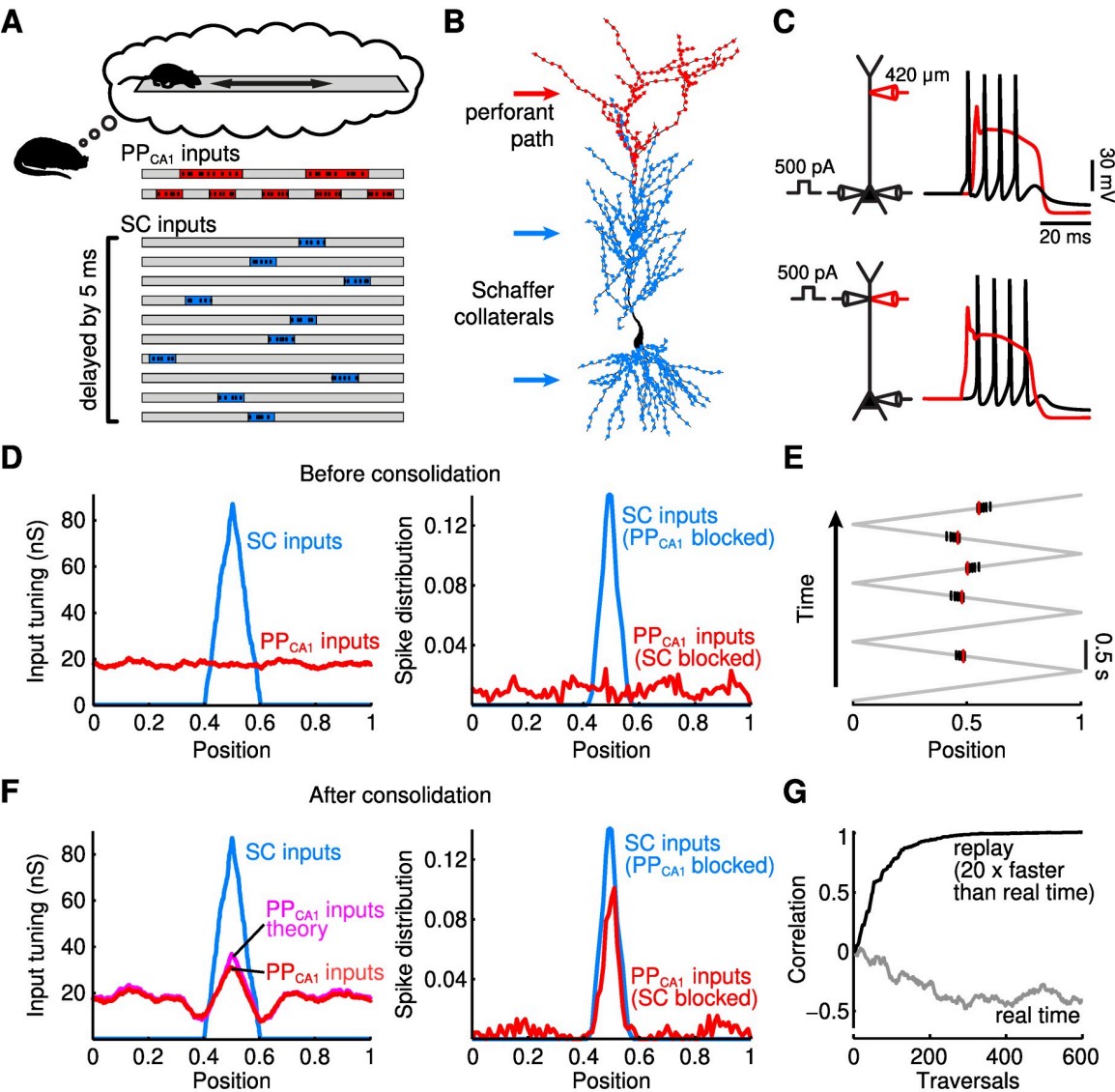

**Fig 3. Consolidation of spatial representations.** (A) Replay of PP$_{CA1}$ and SC activity during sleep. 500 PP$_{CA1}$ inputs and 2500 SC inputs are spatially tuned on a linear track with periodic grid fields (top, red) and place fields (bottom, blue). Spiking activities are independent Poisson processes (10 spikes/s) inside place/grid fields, otherwise silent. SC activity is delayed by 5 ms. (B) Multi-compartmental model of a reconstructed CA1 pyramidal neuron (see Methods). PP$_{CA1}$ and SC inputs project to distal apical tuft dendrites (red dots) and proximal apical and basal dendrites (blue dots). (C) Active neuron properties. Top: somatic sodium spike (black) propagates to the distal tuft and initiates a dendritic calcium spike (red) and further sodium spikes. Bottom: dendritic calcium spike leads to bursts of somatic spikes. (D) Spatial tuning before consolidation. SC provides place field-tuned input to the CA1 cell (left, blue), which yields spatially tuned spiking activity (right, blue); PP$_{CA1}$ input is not spatially tuned (left, red), and (alone) triggers low and untuned spiking activity (right, red). (E) Somatic and dendritic activity during consolidation. During replay, SC input generates backpropagating sodium spikes (black vertical lines) that generate dendritic calcium spikes (red). (F) After consolidation. Spatial tuning is consolidated from the indirect SC pathway into the direct PP$_{CA1}$ pathway. Left: spatial tuning of total PP$_{CA1}$ input (red) approaches theoretically derived PP$_{CA1}$ input tuning (magenta; see Methods). Right: CA1 output is place field-tuned through either SC or PP$_{CA1}$ input alone. (G) Evolution of correlation between actual and optimal PP$_{CA1}$ input tuning (see F) for replay speeds corresponding to hippocampal replay events (black) and real-time physical motion (grey). Position in D, E, and F normalized to [0, 1].

(Fig 3B) that was endowed with active ion channels supporting backpropagating action potentials and dendritic calcium spikes (Fig 3C, Methods).

The use of spatial representations in the input pathways allows us to consider simple forms of memories in a navigational context in which a given location on a linear track is associated

with the activity of a given CA1 cell. Effectively, such an association generates a CA1 place cell. In line with the PPT, we assumed that the spatial selectivity of this CA1 place cell is initially determined solely by the indirect pathway via the SC, i.e., by place cell input from CA3. The goal of systems memory consolidation is then to transfer this spatial association to the direct input, which reaches the CA1 cell via the $PP_{CA1}$ derived from grid cells in EC. In other words, place-cell input should supervise grid-cell input to develop a place-cell tuning. Note that we use the spatial setup primarily as an illustration of the theory. We do not make claims regarding the temporal development of CA1 place cells *in vivo*, which is not fully understood [49–51].

SC place field inputs were modelled by synapses that were active only in a small region of the track, whereas individual $PP_{CA1}$ grid cell inputs were active in multiple, evenly spaced regions along the track (Fig 3A). In terms of the theory, the cue representation in the two pathways is now different, but correlated, because the same location is encoded. The SC and $PP_{CA1}$ inputs projected to proximal and distal dendrites, respectively (Fig 3B, [52]). Synapses were initialized such that the SC input conductances were spatially tuned and resulted in place field-like activity in the CA1 cell while the $PP_{CA1}$ input had no spatial tuning (Fig 3D).

During consolidation, SC and $PP_{CA1}$ input to the CA1 cell consisted of replays of previously encountered sequences of locations [13, 14], with a replay speed 20 times faster than physical motion [13]. During replay, the SC input led to somatic spikes, which in turn triggered back-propagating action potentials that caused calcium spikes in the distal dendrites where the $PP_{CA1}$ synapses arrive (Fig 3C and 3E, [53]). Through synaptic plasticity, $PP_{CA1}$ synapses active in the place field of the neuron were potentiated. Over time, the $PP_{CA1}$ input adopted the spatial tuning of the SC input (Fig 3F, left) and reproduced the original SC-induced place field output (Fig 3F, right) with high correlation (Fig 3G). The fact that the spatial tuning of two inputs is not perfectly matched does not contradict with theoretical results, which merely state that the direct input should attain the best possible linear approximation of the indirect pathway. In the present setting, this approximation is bounded by the finite range of frequencies of the entorhinal inputs (in analogy to reconstructing a high-frequency signal, e.g. a narrow peak, with a finite set of Fourier components), which causes the ringing next to the target peak in Fig 3F (left). In summary, the PPT mechanism therefore consolidated associations even though the spatial representations in the two pathways differed and although the two pathways targeted different neuronal compartments with different numbers of synapses in the CA1 neuron with complex morphology.

The theory also predicts that consolidation is most effective when the correlation time in the input is matched to the time scale of STDP (Fig 2B). In line with this prediction, consolidation failed when replay speed was reduced to that of physical motion (Fig 3G) because the time scale of rate changes in place and grid cell activity is then much longer than the delay between the two pathways and the time scale of STDP (Fig 2C). Accelerated replay during sleep [13] hence supports systems memory consolidation within the PPT by aligning the time scales of neural activity and synaptic plasticity [54], and this alignment is similar to the effect of phase precession during memory acquisition [55].

Finally, we note that the theoretical analysis relied on a separability assumption for the statistics in the two pathways; cf. Eqs (4) and (5). This condition is not fulfilled for sequence replay during consolidation because the time-delayed covariance of different place cells depends on the relative spatial location of their place fields; such correlations are non-separable even for slower replay or during memory acquisition with real-time physical motion. The observation that consolidation was successful nevertheless illustrates that the separability assumption does not need to be fulfilled for the PPT to achieve a successful consolidation.

## Consolidation of place-object associations in multiple hippocampal stages

Ultimately, to consolidate memories into neocortex, they have to move beyond the $PP_{CA1}$. Notably, the $PP_{CA1}$ is itself part of an indirect pathway from EC to the subiculum (SUB) that is shortcut by a direct connection from EC to SUB (referenced as $PP_{SUB}$; Fig 4A, left; [29]). This suggests that the PPT can be reiterated to further consolidate memories from the $PP_{CA1}$ to the $PP_{SUB}$ and beyond.

To illustrate this idea, we considered a standard paradigm for memory research in rodents: the Morris water maze [56]. In the water maze, the rodent needs to find a submerged platform (object), i.e., it must store an object-place association. Thus this paradigm requires neural representations of objects (such as the submerged platform) and places. We hence constructed a model in which subregions of the hippocampal formation included neurons that encode places and neurons that encoded the identity of objects (Fig 4A, right).

For simplicity and computational efficiency we switched to a rate-based neuron model (Methods). An object was chosen from a set of 128 different objects and placed in a circular open field environment (Fig 4B, top). As motivated by experiments [32–34], we implemented

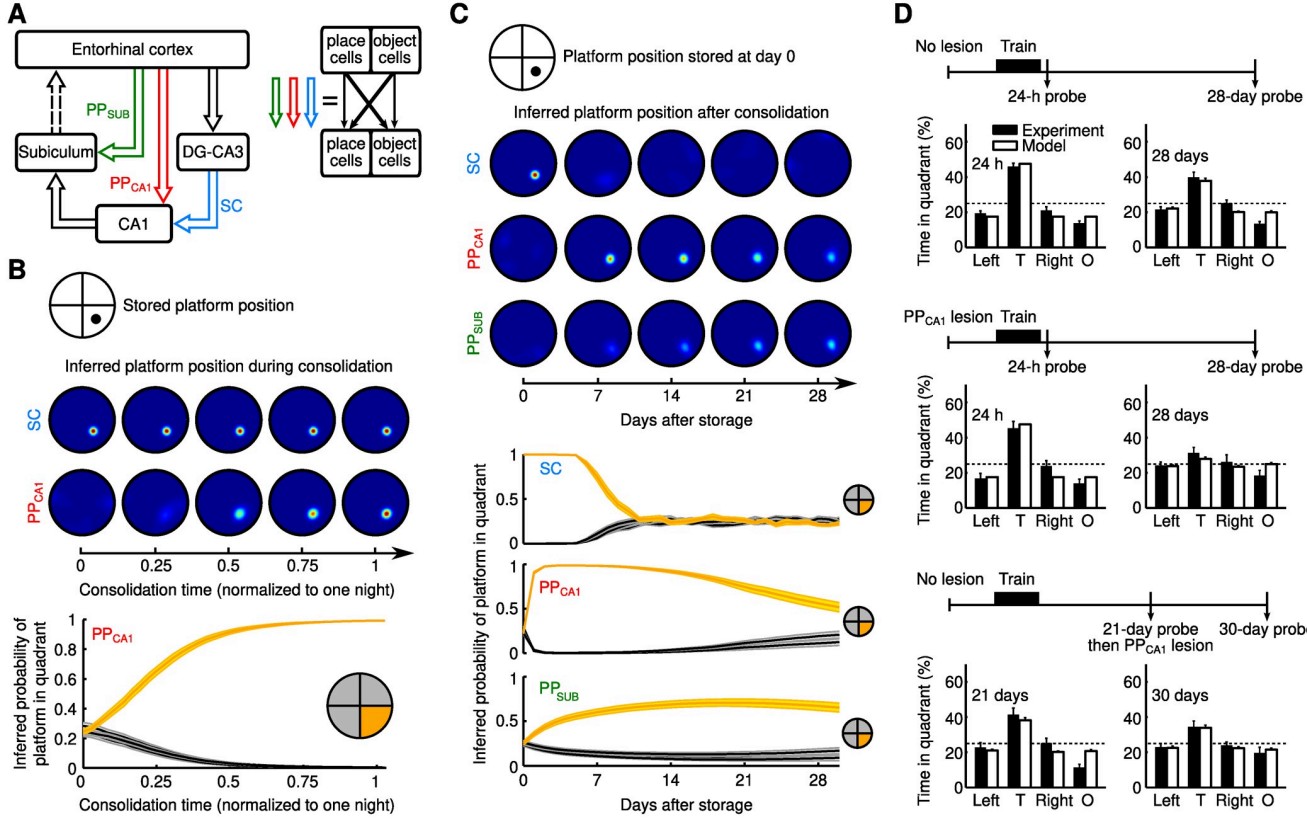

**Fig 4. Consolidation of place-object associations in multiple hippocampal stages.** (A) Structure of the extended model. $PP_{SUB}$: perforant path to the subiculum. Each area (EC, DG-CA3, CA1, SUB) contains object-coding and place-coding populations. Open arrows: all-to-all connections between these areas. (B) Decoding of consolidated associations. Top: The location of a platform in a circular environment is stored as an object-place association in the SC (thick diagonal arrows in A, right). Middle: Platform position probability maps given the platform object cue, inferred from the CA1 output resulting from SC or $PP_{CA1}$ alone, at different times during consolidation (see section "Consolidation of place-object associations in multiple hippocampal stages" in Methods). Bottom: Platform-in-quadrant probabilities (±SEM) given $PP_{CA1}$ input alone during consolidation. Quadrant with correct platform position (target quadrant) in orange. (C) Consolidation from SC to $PP_{CA1}$ and to $PP_{SUB}$ over four weeks. Each day, a new association is first stored in SC and then partially consolidated. An association on day 0 is monitored in SC, $PP_{CA1}$, and $PP_{SUB}$. Panels as in B. (D) Effects of $PP_{CA1}$ lesions on memory consolidation, model and experiment (data with permission from [27]). Histograms of time (±SEM) spent in quadrants at different delays after memory acquisition ("probe"). Dashed lines at 25% are chance levels. T: target quadrant; Left, Right: adjacent quadrants; O: opposite quadrant. Top: Control without lesion. Middle: Lesion before memory acquisition. Bottom: Lesion 21 days after memory acquisition.

object-to-place associations in our model by enhancing, as before, synaptic connections in the SC, but now between object-encoding neurons in CA3 and place-specific neurons in CA1 (Fig 4A, right). Here, we did not consider place-to-object associations. These are less relevant for the water maze task, where the task is to recall the location of a given object—the platform— rather than to recall which object was encountered at a given location. We tested object-to-place associations stored in the SC by activating the object representation in EC—as a memory cue—and determining the activities in CA1, triggered by the SC alone. From these activities we inferred a spatial probability map of the recalled object location (Fig 4B; Methods).

We first stored a single object-place association in the SC. During a subsequent consolidation cycle—representing one night—place and object representations in EC were then randomly and independently activated. Consistent with our previous results, the object-place association was gradually consolidated from the SC to the $PP_{CA1}$: after one night of consolidation, the correct spatial probability map of an object location was inferrable from CA1 activity triggered by the $PP_{CA1}$ alone (Fig 4B).

To track the consolidation process over longer times, we assumed that a new random object-place association is stored in the SC every day. This caused a decay of previous SC memory traces due to interference with newly stored associations (Fig 4C, [57, 58]). During the night following each day, associations in the SC were partially consolidated into the $PP_{CA1}$, such that the consolidated association could be decoded from the $PP_{CA1}$ after a single night, but previously consolidated associations were not entirely overwritten. As a result, object-place associations were maintained in the $PP_{CA1}$ for longer periods than in the SC, thus extending their memory lifetime (Fig 4C). Eventually, a given $PP_{CA1}$ memory trace would also degrade as new interfering memories from the SC are consolidated. However, as noted above, the $PP_{CA1}$ itself is part of an indirect pathway from EC to the SUB, for which there is in turn a parallel, direct perforant pathway $PP_{SUB}$. The association in the $PP_{CA1}$ (and SC) could therefore, in turn, be partially consolidated into the $PP_{SUB}$, further extending memory lifetime (Fig 4C). Note that the extension of memory lifetime is supported in the model by a reduced plasticity (i.e. halved learning rate in Eq (33)) in $PP_{SUB}$ compared to $PP_{CA1}$.

The model suggests that the $PP_{CA1}$ serves as a transient memory buffer that mediates a further consolidation into additional shortcut pathways downstream. This hypothesis is supported by navigation studies in rats. Using $PP_{CA1}$ lesions, Remondes and Schuman [27] have shown that the $PP_{CA1}$ is not required for the original acquisition of spatial memories, but that it is critically involved in their long-term maintenance. However, lesioning the $PP_{CA1}$ 21 days after acquiring a memory did not disrupt spatial memories, suggesting that the $PP_{CA1}$ is not the final storage site (Fig 4D) and further supporting the idea that the $PP_{CA1}$ is important to enable a transition from short-term to long-term memories.

To test whether our model could reproduce these experimental results, we simulated $PP_{CA1}$ lesions either before the acquisition of an object-place association or 21 days later. Assuming that the rat's spatial exploration is determined by the probability map of the object location [59], the model provided predictions for the time spent in different quadrants of the environment, which were in quantitative agreement with the data for all experimental conditions (Fig 4D). Our model thus suggests that a hierarchical reiteration of parallel shortcuts—the central circuit motif of the PPT—could explain these experiments.

Similar to lesioning the $PP_{CA1}$, we predict that lesioning $PP_{SUB}$ also has an impact on memory consolidation: $PP_{SUB}$ should act as a transient memory buffer but on a longer timescale than $PP_{CA1}$. In general, lesioning a pathway with a set of synapses that cover a specific range of time scales such that there is a "gap" should result in an impairment of consolidation if the lesion is done before the memory has "moved on". To illustrate this idea in more detail, we study in the next section a model with many stages in a hierarchy.

## Consolidation from hippocampus into neocortex by a hierarchical nesting of consolidation circuits

Given that shortcut connections are widespread throughout the brain [25, 60, 61], we next hypothesized that a reiteration of the PPT can also achieve systems consolidation from hippocampus into neocortex. To test this hypothesis, we studied a network model (Fig 5A), in which the hippocampus (now simplified to a single area) receives input from a hierarchy of cortical areas, representing, e.g., a sensory system. It provides output to a different hierarchy of areas, representing, e.g., the motor system or another sensory system.

The network also contained shortcut connections that bypassed the hippocampus. As in the previous section, new memories were stored in the hippocampus but not in any other indirect connection in the hierarchy. The repeated storage of new memories every day leads to a decay of previously stored hippocampal memories. But memories are also consolidated by Hebbian plasticity in parallel pathways; for details, see Methods.

Tracing a specific memory over time revealed a gradual consolidation into the cortical shortcut connections, forming a "memory wave" [10] that travels from hippocampus into neocortex (Fig 5B). By exponentially decreasing the shortcut learning rate with distance from the hippocampus, a power-law decay of memories can be observed in the union of all shortcuts, e.g., by reading out the shortcut with the strongest memory trace at any moment in time (Fig 5B). This observation is in line with a rich history of psychological studies on the mathematical

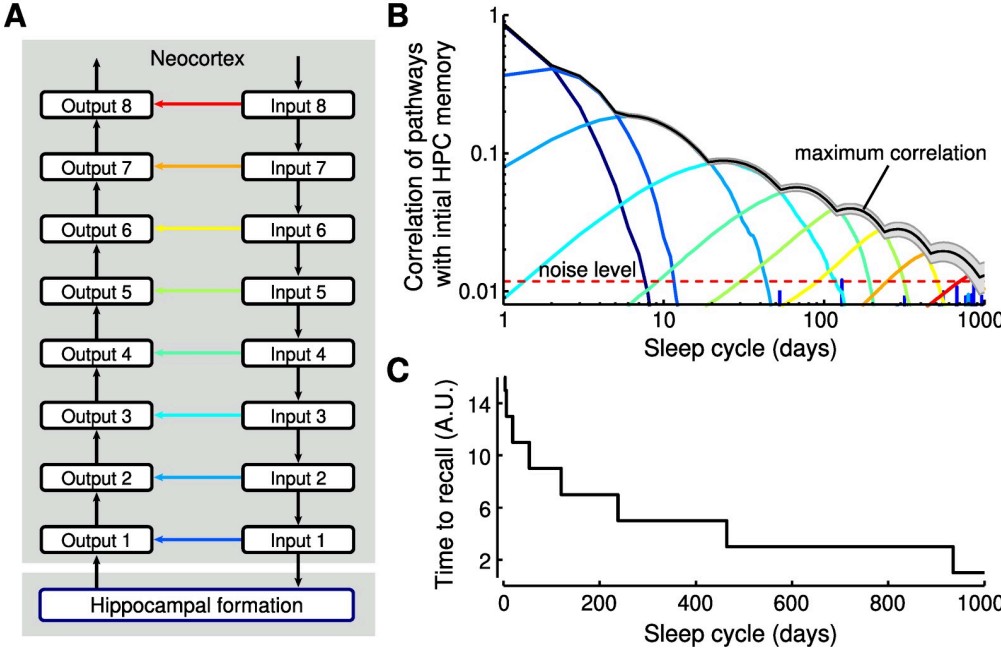

**Fig 5. Consolidation from hippocampus into neocortex by hierarchical nesting of consolidation circuits.** (A) Schematic of the hierarchical model. The hippocampal formation (HPC) is connected to cortical input circuit 1 and output circuit 1. Increasing numbers indicate circuits further from the HPC and closer to the sensory/motor periphery. Each direct connection at one level (e.g., dark blue arrow between input 1 and output 1) is part of the indirect pathway of the next level (e.g., for pathways from input 2 to output 2). Learning rates of the direct connections decrease exponentially with increasing level (i.e., from blue to red). (B) Memories gradually propagate to circuits more distant from the HPC. The correlation of the initial HPC weights with the direct pathways is shown as a function of time and reveals a memory wave from HPC into neocortex. The maximum of the output circuits follows approximately a power-law (black curve). Noise level indicates chance-level correlations between pathways. (C) Consolidated memories yield faster responses (from sensory periphery, e.g., Input 8, to system output) because these memories are stored in increasingly shorter synaptic pathways.

shape of forgetting curves [28]. Note that for the readout we tried to make as few assumptions as possible by letting all pathways contribute on an equal footing. Taking the maximum over the pathways (as well as the mean) generates a power law. Notably, we achieved memory retention times of years through only a small number (∼5) of iterations of the PPT. Finally, we found that memory retrieval accelerates during consolidation (Fig 5C), in line with consolidation studies for motor skills [62]. In our consolidation model, the time to recall decreases because the path from peripheral input to output becomes shorter through the use of more direct (peripheral) shortcut connections (Fig 5A and 5B).

The predicted consolidation-mediated decrease of the time to recall critically depends on the utilized plasticity rule (STDP), which uses timing of input and output of neurons, and on our assumption that memories are initially acquired in an indirect pathway with a longer delay than direct pathways. While this assumption is reasonable for declarative memories that are initially stored in the hippocampus and then consolidated in sensory or motor areas towards neocortex, the underlying computational reasons for such a strategy are unknown. The strategy of the initial storage of memories in a pathway with a longer transmission delay could be related to the Complementary Learning Systems Theory (CLST) [11, 63] if the initial storage needs some preprocessing, e.g., to achieve representations that are suited for one-trial learning, e.g. population-sparse representations [9]. In general, our results do not imply that the reduction of delay is a central goal of systems memory consolidation or that it is even necessary. Reduction of delay may, however, be a nice side effect of systems memory consolidation with timing-based plasticity rules [64]. And such a reduction of delay does not need to be restricted to declarative memories but also could apply to, e.g., motor skill learning or habit formation.

## Discussion

We proposed the parallel pathway theory (PPT) as a mechanistic basis for systems memory consolidation. This theory relies on two abundant features in the nervous system: parallel shortcut connections between brain areas and Hebbian plasticity. A mathematical analysis suggests that STDP in a direct pathway achieves consolidation by implementing a linear regression that approximates the input-output mapping of an indirect pathway by that of the direct pathway. We applied the PPT to hippocampus-dependent memories and showed that the proposed mechanism can transfer memory associations across parallel synaptic pathways. This transfer is robust to different representations in those pathways and requires only weak correlations. Our results are in quantitative agreement with lesion studies of the perforant path in rodents [27] and are able to reproduce forgetting curves that follow a power-law as observed in humans [28].

### Theory requirements and predictions

In addition to the anatomical motif of shortcut connections and Hebbian synaptic plasticity, the parallel pathway theory relies on four further requirements during the consolidation phase, which can also be considered as model predictions.

(1). Temporal correlations between the inputs from the two input pathways are necessary during consolidation, and these correlations should be similar to the ones during storage and recall. For example, a consolidation from hippocampus into neocortex would require correlations between cortical and hippocampal activity, as reported in [65]. Similarly, a consolidation of spatial memories within the hippocampal formation (including the medial entorhinal cortex, MEC) during replay would require correlations between activity in MEC and hippocampus; in particular, the same locations should be replayed, but

represented by grid cells in MEC and by place cells in CA3 and CA1, as in Fig 3. A significant but weak correlation between the superficial layers of MEC (which provides input to the hippocampus) and CA1 was indeed observed [45]. Furthermore, pyramidal cells in the superficial layer III (projecting to CA1, "direct path") and stellate cells in the superficial layer II (projecting to DG, which projects to CA3, "indirect path") are expected to be correlated due to a strong excitatory feedforward projection from pyramids to stellates [66]; reviewed in [51]. Coordinated grid and place cell replay was also observed in [67] but there CA1 and deep layers of MEC (which receives the hippocampal output) were studied.

(2). The direct pathway should be plastic during consolidation, while the stored associations in the indirect path remain sufficiently stable (in contrast to the model in [24]). In practice, this requires the degree of plasticity to differ between periods of storage and consolidation (e.g., due to neuromodulation [68, 69]), in a potentially pathway-dependent manner. In other words, the requirement is that the content of a memory should not be altered much while creating a backup.

(3). Plasticity in the shortcut pathway should be driven by a teaching signal from the indirect pathway. This can be achieved by STDP in combination with longer transmission delays in the indirect pathway, as suggested here, but other neural implementations of supervised learning may be equally suitable [42, 43, 70].

(4). Within the present framework, a systematic decrease in learning rates within the consolidation hierarchy (Fig 5) is needed to achieve memory lifetimes on the order of years. That is, synapses involved in later stages of consolidation should be less plastic during consolidation periods such as sleep, as also suggested by [10] and [24]. Furthermore, Roxin and Fusi elegantly showed in [10] that a multistage memory system confers an advantage (in terms of memory lifetime, memory capacity, and initial signal-to-noise ratio) compared to a homogeneous memory system with the same number of synapses, which provides a fundamental computational reason for the existence of a memory consolidation processes at the systems level. However, to be able to exploit this advantage, an efficient mechanism to transfer memories across stages is necessary. The proposed PPT explains how memories can be transferred in a biologically plausible way in a multistage memory system. Conceptually related to models of systems-memory consolidation with a systematic decrease in learning rates across a hierarchy of networks are models of synaptic memory consolidation with complex synapses that can assume many different states and a decrease of plasticity across a hierarchy of states. In such models of synaptic memory consolidation, also a power-law forgetting has been achieved [8, 71]. Synaptic and systems memory consolidation models are different but not mutually exclusive.

## What limits systems memory consolidation?

Our account of systems memory consolidation explains how memories are re-organized and transferred across brain regions. However, certain forms of episodic memory remain hippocampus-dependent throughout life [21].

In the context of the present model, this restriction could result from different factors. The PPT simplifies memory engrams by replacing multisynaptic by monosynaptic connections whenever possible. However, a shortcut pathway may not be present anatomically, or it may not host an appropriate representation for a given cue-response association in question. For example, it may be difficult to consolidate a complex visual object detection task into a

shortcut from primary visual cortex (V1) to a decision area because the low-level representation of the visual cue in V1 may not allow it [72, 73]. The same applies to tasks that require a mixed selectivity of neural responses [74]. Such tasks cannot be fully consolidated into shortcuts with simpler representations of cues and/or responses that do not allow a linear separation of the associations. On the basis of similar arguments, early work suggested that the hippocampus could be critical for learning tasks that are not linearly separable [75].

Within the present framework, the consolidated memory is in essence a linear approximation of the original cue-response association, as indicated in the theoretical analysis around Eqs (7) and (8). The resulting simplification of the memory content could underlie the commonly observed semantization of memories and the loss of episodic detail [20, 21]. Such a semantization could already occur in the earliest shortcut connections [76], but could also gradually progress in a multi-stage consolidation process.

### Relation to phenomenological models of systems consolidation

The basic mechanism of our framework explains memory transfer between brain regions, which is in line with the Standard Consolidation Theory (SCT) [11, 19]. Our theoretical framework is closely related to the Complementary Learning Systems Theory (CLST) [11, 63], which posits that slow and interleaved cortical learning is necessary to avoid catastrophic interference of new memory items with older memories [77]. In our model, later—presumably neocortical—shortcut connections have lower learning rates to achieve longer memory retention times. Interleaved learning could be achieved by interleaved replay [78–80] during consolidation. Thereby, the results of CLST can be directly applied to learning in shortcuts in our model, such as the rapid neocortical consolidation of new memories that are in line with a previously learned schema [17, 63, 81].

Limitations of memory transfer between brain regions—as discussed above—can impair the consolidation process, resulting in memories that remain hippocampus-dependent throughout life. Hence, our theoretical framework is also in agreement with the Multiple Trace Theory (MTT) [16] and the Trace Transformation Theory (TTT) [20, 21]. The MTT postulates that memories are re-encoded in the hippocampus during retrieval, generating multiple traces for the same memory. Our model maintains multiple memory traces in different shortcut pathways, even without a retrieval-based re-encoding. The consolidation mechanism of the PPT, however, could also transfer a specific memory multiple times if it is re-encoded during retrieval. If neocortex extracts statistical regularities from a collection of memories [11], the consolidation of such a repeatedly re-encoded memory could then lead to a gist-like, more semantic version of that memory in neocortex [16, 21, 82], as emphasized by the TTT.

The premise of our model is that memories are actively transferred between brain regions. This premise has recently been subject to debate [83–85], following the suggestion of the Contextual Binding (CB) theory. The CB theory argues that amnesia in lesion studies and replay-like activity can be explained by simultaneous learning in hippocampus and neocortex, together with interference of contextually similar episodic memories [83]. Note, however, that our framework does not exclude a simultaneous encoding in neocortex and hippocampus, which can be combined with active consolidation [1, 86].

Hence, our mechanistic approach is in agreement with and may allow for a unification of several phenomenological theories of systems consolidation.

### Consolidation of non-declarative memories

Given that shortcut connections are widespread throughout the central nervous system [25, 60], the suggested mechanism may also be applicable to the consolidation of non-declarative

memories, e.g., of perceptual [4] and motor skills [5], fear memory [87] or to the transition of goal-directed to habitual behaviour [88].

Several studies have suggested two-pathway models in the context of motor learning [89–92]. In particular, Murray and Escola [92] recently used a two-pathway model to investigate how repeated practice affects future performance and leads to habitual behaviour. While their model does not incorporate an active consolidation mechanism or multiple learning stages, the basic mechanism is the same: A fast learning pathway from cortex to sensorimotor striatum first learns a motor skill and then teaches a slowly learning pathway from thalamus to striatum during subsequent repetition.

## Limitations of the model and future directions

The present work focuses on feedforward networks and local learning rules. Hence, the model cannot address how systems memory consolidation affects the representation of sensory stimuli and forms schemata that facilitate future learning [17, 81] because representation learning typically requires a means of backpropagating information through the system, e.g., by feedback connections [93]. The interaction of synaptic plasticity with recurrent feedback connections generates a high level of dynamical complexity, which is beyond the scope of the present study. Our framework also does not explain reconsolidation, that is, how previously consolidated memories become labile and hippocampus-dependent again through their reactivation [94, 95].

On the mechanistic level, the PPT predicts temporally specific deficits in memory consolidation when relevant shortcut connections are lesioned, that is, a tight link between the anatomical organisation of synaptic pathways and their function for memory. These predictions may be most easily tested in non-mammalian systems, where connectomic data are available [96].

The PPT could provide an inroad to a mechanistic understanding of the transformation of episodic memories into more semantic representations. This could be modelled, e.g, by encoding a collection of episodic memories that share statistical regularities and studying the dynamics of statistical learning and semantisation in the shortcut connections during consolidation. Such future work may allow us to ultimately bridge the gap between memory consolidation on the mechanistic level of synaptic computations and the behavioural level of cognitive function.

## Methods

### Consolidation in a single integrate-and-fire neuron

For the results shown in Fig 1E and 1F we used a single integrate-and-fire model neuron that received excitatory synaptic input. The membrane potential $V(t)$ evolved according to

$$\tau_\mathrm{m} \frac{\mathrm{d}V}{\mathrm{d}t} = V_\mathrm{rest} - V + g_\mathrm{syn}(t)(E_\mathrm{syn} - V)\,, \tag{9}$$

with membrane time constant $\tau_\mathrm{m} = 20$ ms, resting potential $V_\mathrm{rest} = -70$ mV, and synaptic reversal potential $E_\mathrm{syn} = 0$ mV. When the membrane potential reached the threshold $V_\mathrm{thresh} = -54$ mV, the cell produced a spike and the voltage was reset to $-60$ mV during an absolute refractory period of 1.75 ms.

The total synaptic conductance $g_\mathrm{syn}(t)$ in Eq (9) is denoted in units of the leak conductance and thus dimensionless (parameters are taken from [97]). The total synaptic conductance was determined by the sum of 1000 Schaffer collateral (SC) inputs and 1000 perforant path (PP$_\mathrm{CA1}$) inputs. Activation of input $i$ (where $i$ denotes synapse number) leads to a jump $g_i > 0$

in the synaptic conductance:

$$g_{\text{syn}}(t) \rightarrow g_{\text{syn}}(t) + g_i .$$

(10)

All synaptic conductances decay exponentially,

$$\tau_{\text{syn}} \frac{dg_{\text{syn}}}{dt} = -g_{\text{syn}} ,$$

(11)

with synaptic time constant $\tau_{\text{syn}}$ = 5 ms. The PP$_{\text{CA1}}$ inputs were activated by mutually independent Poisson processes with a mean rate of 10 spikes/s. The activity patterns of the SC fibers were identical to those of the PP$_{\text{CA1}}$ fibers but were delayed by 5 ms.

The synaptic peak conductances or weights, $g_i$, were either set to a fixed value or were determined by additive STDP [98]. A single pair of a presynaptic spike (at time $t_{\text{pre}}$) and a postsynaptic spike (at time $t_{\text{post}}$) with time difference $\Delta t \equiv t_{\text{pre}} - t_{\text{post}}$ induced a modification of the synaptic weight $\Delta g_i$ according to

$$\Delta g_i = L(\Delta t) = \begin{cases} +A^+ \exp(\Delta t / \tau_{\text{STDP}}) & \text{if } \Delta t < 0, \\ -A^- \exp(-\Delta t / \tau_{\text{STDP}}) & \text{if } \Delta t \geq 0, \end{cases}$$

(12)

with $\tau_{\text{STDP}}$ = 20 ms. $L(\Delta t)$ is the learning window of STDP [98]. Hard upper and lower bounds were imposed on the synaptic weights, such that $0 \leq g_i \leq \bar{g}_{\text{max}}$ for all $i$, where the dimensionless maximum synaptic weight was $\bar{g}_{\text{max}} = 0.006$. Parameters $A^+ = \eta \cdot \bar{g}_{\text{max}}$ and $A^- = 1.05 \cdot A^+$ with $\eta = 0.005$ determine the maximum amounts of LTP and LTD, respectively.

Synaptic weights were initialized to form a bimodal distribution, such that it agrees with the steady state weight distribution resulting from additive STDP, when presynaptic input consists of uncorrelated Poisson spike trains [98]. Specifically, half the weights were sampled from an exponential distribution with mean $0.05 \cdot \bar{g}_{\text{max}}$, the other half as $\bar{g}_{\text{max}}$ minus that same exponential distribution.

The dynamics were integrated numerically using the forward Euler method, with an integration time step of 0.1 ms.

## Consolidation of spatial representations in a multi-compartment neuron model

The results presented in Fig 3C–3G relied on numerical simulations of a conductance-based compartmental model of a reconstructed CA1 pyramidal cell (cell n128 from [99]). Passive cell properties were defined by the membrane resistance $R_{\text{m}}$ = 30 k$\Omega$ cm$^2$ with reversal potential $E_{\text{L}}$ = −70 mV, intracellular resistivity $R_{\text{i}}$ = 150 $\Omega$cm, and membrane capacitance $C_{\text{m}}$ = $0.75\mu$F/cm$^2$. Dendrites were discretized into compartments with length smaller than 0.1 times the frequency-dependent passive space constant at 100 Hz. Three types of voltage-dependent currents and one calcium-dependent current, all from [100], were distributed over the soma and dendrites. Gating dynamics of the currents evolved according to standard first-order ordinary differential equations. The steady state (in)activation functions $x_\infty$ and voltage-dependent time constants $\tau_\infty$ for each gating variable (i.e., $x = m, h, n$; see below) were calculated from a first-order reaction scheme with forward rate $\alpha_x$ and backward rate $\beta_x$ according to $x_\infty(V) = \alpha_x(V)/(\alpha_x(V) + \beta_x(V))$ and $\tau_x(V) = 1/(\alpha_x(V) + \beta_x(V))$ where $V$ was the membrane potential. All used current densities and time constants were selected for a temperature of 37°C (see [100]).

A fast sodium current, $I_{Na}$, was distributed throughout the soma ($\bar{g}_{Na} = 130$ pS/$\mu$m$^2$) and dendrites ($\bar{g}_{Na} = 260$ pS/$\mu$m$^2$), except from the distal apical dendritic tuft,

$$I_{Na} = \bar{g}_{Na} m^3 h (V - E_{Na}) \,, \tag{13}$$

with reversal potential $E_{Na} = 60$ mV. The dynamics of activation gating variable $m$ and inactivation gating variable $h$ were characterized by

$$
\begin{aligned}
\alpha_m &= -0.584 \frac{V + 30}{e^{-(V+30)/9} - 1} \\
\beta_m &= 0.398 \frac{V + 30}{e^{(V+30)/9} - 1} \\
\alpha_h &= -0.077 \frac{V + 45}{e^{-(V+45)/5} - 1} \\
\beta_h &= 0.0292 \frac{V + 70}{e^{(V+70)/5} - 1} \,.
\end{aligned}
\tag{14}
$$

Here and in the following, we dropped units for simplicity, assuming that the membrane potential $V$ is given in units of mV.

The steady-state inactivation function was defined directly as

$$h_\infty = \frac{1}{1 + e^{(V+60)/6.2}} \,. \tag{15}$$

A fast potassium current, $I_{Kv}$, was present in the soma ($\bar{g}_{Kv} = 95$ pS/$\mu$m$^2$) and throughout the dendrites ($\bar{g}_{Kv} = 190$ pS/$\mu$m$^2$),

$$I_{Kv} = \bar{g}_{Kv} n (V - E_K) \,, \tag{16}$$

with reversal potential $E_K = -90$ mV and with activation gating variable $n$ characterized by

$$
\begin{aligned}
\alpha_n &= -0.064 \frac{V - 25}{e^{-(V-25)/9} - 1} \\
\beta_n &= 0.0064 \frac{V - 25}{e^{(V-25)/9} - 1} \,.
\end{aligned}
\tag{17}
$$

A high-voltage activated calcium current, $I_{Ca}$, was distributed throughout the apical dendrites ($\bar{g}_{Ca} = 30$ pS/$\mu$m$^2$) with an increased density ($\bar{g}_{Ca} = 35$ pS/$\mu$m$^2$) for dendrites distal from the main apical dendrite's bifurcation,

$$I_{Ca} = \bar{g}_{Ca} m^2 h (V - E_{Ca}) \,, \tag{18}$$

with reversal potential $E_{Ca} = 140$ mV and with activation gating variable $m$ and inactivation gating variable $h$ characterized by

$$
\begin{aligned}
\alpha_m &= -0.177 \frac{V + 27}{e^{-(V+27)/3.8} - 1} \\
\beta_m &= 3.02 \, e^{-(V+75)/17} \\
\alpha_h &= 4.89 \cdot 10^{-4} \, e^{-(V+13)/50} \\
\beta_h &= \frac{0.0071}{e^{-(V+15)/28} + 1} \,.
\end{aligned}
\tag{19}
$$

A calcium-dependent potassium current, $I_{KCa}$, was similarly distributed throughout the apical dendrites ($\bar{g}_{KCa} = 30$ pS/$\mu$m$^2$) with an increased density ($\bar{g}_{KCa} = 35$ pS/$\mu$m$^2$) beyond the main bifurcation of the apical dendrite,

$$I_{KCa} = \bar{g}_{KCa} n (V - E_K) \,, \tag{20}$$

with activation gating variable $n$ characterized by

$$\alpha_n = 0.032([Ca^{2+}]_i)^6$$
$$\beta_n = 0.064 \tag{21}$$

with $[Ca^{2+}]$ in $\mu$M.

Internal calcium concentration in a shell below the membrane surface was computed using entry via $I_{Ca}$ and removal by a first-order pump,

$$\frac{d[Ca^{2+}]_i}{dt} = -\frac{10,000}{2Fd} I_{Ca} + \frac{[Ca^{2+}]_\infty - [Ca^{2+}]_i}{\tau_R} \,, \tag{22}$$

with Faraday constant $F$, depth of shell $d = 0.1$ $\mu$m and with $[Ca^{2+}]_\infty = 0.1 \mu$M, and $\tau_R = 80$ ms. To account for dendritic spines, the membrane capacitance and current densities were doubled throughout the dendrites. An axon was lacking in the cell reconstruction and was added as in [100].

Excitatory synaptic inputs were distributed over the membrane surface. Upon activation of a synapse, the conductance with a reversal potential of 0 mV increased instantaneously and subsequently decayed exponentially with a time constant of 3 ms. The PP$_{CA1}$ provided 500 inputs that were distributed with uniform surface density throughout the distal apical tuft dendrites; the SC provided 2500 inputs, distributed uniformly over basal dendrites and proximal apical dendrites [52].

All inputs were spatially tuned on a 2.5 m long linear track over which the simulated rat walked. The PP$_{CA1}$ inputs showed periodic, grid field-like spatial tuning with periodicity ranging from 2 to 6 grid fields along the entire track with random phase: $G_i(x) = r\mathcal{H}(\cos(2\pi kx + \xi_i))$, where $\mathcal{H}$ is the Heaviside step function, $r$ is the mean firing rate within the grid field, $k$ is the spatial frequency, and $\xi_i$ is the random spatial phase offset for neuron $i$ (for $i = 1, \ldots, 500$). The 2500 SC inputs showed place field-like tuning, having single, 25 cm long place fields distributed uniformly random along the track. When the virtual rat was within the place or grid field of an SC or PP$_{CA1}$ fiber, respectively, the input was activated as an independent Poisson process with a mean rate of $r = 10$ spikes/s. Outside of the place/grid fields the fibers were quiescent. Simulations of the consolidation phase considered replay of the rat walking back and forth along the linear track, with running speeds increased, compared to realistic speeds, by a factor 20 (5 m/s; [13]). SC input activity to the CA1 cell was delayed by 5 ms with respect to the PP$_{CA1}$ input [101], accounting for the extra processing stages involved for information reaching CA1 from the entorhinal cortex through DG and CA3, compared to the direct entorhinal PP$_{CA1}$ input.

The PP$_{CA1}$ and/or SC inputs showed additive STDP, operating in the same manner as defined around Eq (12). Post-synaptic spikes were defined as local voltage crossings of a threshold at −30 mV. The maximum synaptic weight for the SC inputs was 400 pS and 140 pS for the PP$_{CA1}$ inputs.

The reference tuning curve shown in Fig 3F (PP$_{CA1}$ inputs theory) was computed by adding up all grid field tuning functions that had an active field in the SC-encoded spatial position (i.e., halfway along the linear track).

Simulations were carried out with a fixed time step of 25 $\mu$s using the NEURON simulation software [102].

## Consolidation of place-object associations in multiple hippocampal stages

The results related to Fig 4 show the acquisition and consolidation of place-object associations in a hippocampal network model. Every day a virtual animal learns the position of one of many possible objects in a circular open field environment. The simulations show that during a subsequent sleep phase, replay of the hippocampal activity that is associated with runs through this environment allows for the consolidation of the place-object association. We call the imprinting of a new memory and the subsequent memory consolidation phase a consolidation cycle. In the simulations, a place-object association learned at time $t = 0$ is tracked for $N_{\text{cycle}}$ consolidation cycles, i.e., nights after memory acquisition. Between consolidation cycles, the memory in the system is assessed as described below.

**Model architecture.** The model consists of four neuronal layers: entorhinal cortex (EC), dentate gyrus/CA3 (DG-CA3; note that the dentate gyrus is not explicitly included as a separate area), CA1, and the subiculum (SUB). Each layer consists of a population of place-coding cells and a population of object-coding cells. The connectivity is depicted in Fig 4A: EC projects to DG-CA3, which connects to CA1 (through the SC pathway), which in turn connects to the SUB. EC provides also shortcut connections to CA1 ($\text{PP}_{\text{CA1}}$ pathway) and the SUB ($\text{PP}_{\text{SUB}}$ pathway).

The SC, $\text{PP}_{\text{CA1}}$, and $\text{PP}_{\text{SUB}}$ pathways consist of four different connection types among populations of neurons that represent either place or object: (i) from object (populations) to object (populations), (ii) from place to place, (iii) from object to place, and (iv) from place to object. For simplicity, the pathway from CA1 to the SUB consists only of place-to-place and object-to-object connections, because we never store object-place or place-object associations in this pathway. The pathway from EC to DG-CA3 was not explicitly modelled. Instead, we assumed that the same location (of the virtual animal) is represented in both areas, but with a grid cell code and a place cell code, respectively. We assumed that all connections have the same transmission delay, which is equal to one time step $D = \Delta T = 5$ ms in the simulation (see Table 1 for parameter values). In practice, this meant that the activities in the SC pathway and the connection from CA1 to the SUB each had a transmission delay $D$ relative to the activities in the connections from EC to DG/CA1 and from EC to SUB.

Activities of neurons in each layer were described as firing rates and were determined by a linear model,

$$\mathbf{y}_{\text{CA1}}(t) = \mathbf{W}_{\text{PP-CA1}}^{\mathsf{T}}(t)\,\mathbf{x}_{\text{EC}}(t) + \mathbf{V}_{\text{SC}}^{\mathsf{T}}\,\mathbf{x}_{\text{CA3}}(t - D), \tag{23}$$

$$\mathbf{y}_{\text{SUB}}(t) = \mathbf{W}_{\text{PP-SUB}}^{\mathsf{T}}(t)\,\mathbf{x}_{\text{EC}}(t) + \mathbf{V}_{\text{CA1-SUB}}^{\mathsf{T}}\,\mathbf{y}_{\text{CA1}}(t - D), \tag{24}$$

where $\mathbf{x}_{\text{EC}}(t)$ and $\mathbf{x}_{\text{CA3}}(t)$ are the activities in the input layers EC and DG-CA3, respectively, and $\mathbf{y}_{\text{CA1}}(t)$ and $\mathbf{y}_{\text{SUB}}(t)$ represent the activities in the output layers CA1 and SUB, respectively. Time is denoted by $t$. The symbols $\mathbf{W}_{\text{PP-CA1}}$ and $\mathbf{W}_{\text{PP-SUB}}$ denote the weight matrices of the pathways from EC to CA1 and from EC to SUB, respectively. The matrices $\mathbf{V}_{\text{SC}}$ and $\mathbf{V}_{\text{CA1-SUB}}$ summarise the weights from DG-CA3 to CA1 and from CA1 to SUB, respectively, which mediate the transmission delay $D$. Eqs (23) and (24) are identical in structure to Eq (1) except that now the output is a vector (and not a scalar) and the synaptic weights are a matrix (and not a vector).

**Table 1. Parameters for simulations shown in Fig 4.**

| | | |
|---|---|---|
| $N_{\text{cycle}}$ | 31 | number of consolidation cycles |
| $T_c$ | 150 s | consolidation time per sleep cycle |
| $\Delta T$ | 5 ms | integration time step |
| $N$ | 256 | neurons per place- or object-coding population |
| $N_{\text{object}}$ | 128 | number of different objects |
| $r_{\text{max}}$ | 10 spikes/s | maximum output firing rate |
| $\sigma$ | 0.1 | size of place field standard deviation |
| $D$ | 5 ms | transmission delay |
| $w_{\text{SC}}^{\text{id}}$ | $\frac{1}{4}$ | weight between object-object and place-place coding cells in DG-CA3 and CA1 |
| $w_{\text{CA1-SUB}}^{\text{id}}$ | $\frac{1}{2}$ | weight between object-object and place-place coding cells in CA1 and SUB |
| $\lambda_{\text{SC}}$ | 0.6 | relative strength of new place-object association in $\mathbf{V}_{\text{SC}}$ |
| $N_{\text{mem}}$ | 125 | number of associations stored to initialize $\mathbf{V}_{\text{SC}}$ |
| $w_{\text{max}}$ | $\frac{1}{N}$ | maximum weight values for $\mathbf{W}_{\text{PP-CA1}}$ and $\mathbf{W}_{\text{PP-SUB}}$ |
| $w_{\text{init}}^{\text{max}}$ | $\frac{1}{10} \cdot w_{\text{max}}$ | maximum initial weight values for $\mathbf{W}_{\text{PP-CA1}}$ and $\mathbf{W}_{\text{PP-SUB}}$ |
| $A_{\text{PP-CA1}}^{+}$ | $0.05 \cdot w_{\text{max}}$ | height of potentiating learning window for $\mathbf{W}_{\text{PP-CA1}}$ |
| $A_{\text{PP-CA1}}^{-}$ | $-1.00025 \cdot A_{\text{PP-CA1}}^{+}$ | height of depressing learning window for $\mathbf{W}_{\text{PP-CA1}}$ |
| $A_{\text{PP-SUB}}^{+}$ | $0.5 \cdot A_{\text{PP-CA1}}^{+}$ | height of potentiating learning window for $\mathbf{W}_{\text{PP-SUB}}$ |
| $A_{\text{PP-SUB}}^{-}$ | $0.5 \cdot A_{\text{PP-CA1}}^{-}$ | height of depressing learning window for $\mathbf{W}_{\text{PP-SUB}}$ |
| $\tau_{\text{STDP}}$ | 20 ms | time constants of learning window |
| $N_{\text{equi}}$ | 10 | equilibration sleep phases run before the simulation starts |
| $\sigma_{\text{noise}}$ | 4.8 | noise level assumed for place inference |

As already mentioned above, each neuron in a layer is assumed to primarily encode either place or object information (see Fig 4A). To simplify the mathematical analysis, we turn to a notation where we write a layer's activity vector $\mathbf{z}$ (where $\mathbf{z} = \mathbf{x}_{\text{EC}}$, $\mathbf{x}_{\text{CA3}}$, $\mathbf{y}_{\text{CA1}}$, or $\mathbf{y}_{\text{SUB}}$) as a concatenation of place and object vectors:

$$\mathbf{z} = \begin{bmatrix} \mathbf{z}^{\text{place}} \\ \mathbf{z}^{\text{object}} \end{bmatrix}, \tag{25}$$

where the number of place- and object-coding cells is identical, $\dim(\mathbf{z}^{\text{place}}) = \dim(\mathbf{z}^{\text{object}}) = N$, hence $\dim(\mathbf{z}) = 2N$. Correspondingly, the weight matrices $\mathbf{M}$ (where $\mathbf{M} = \mathbf{W}_{\text{PP-CA1}}$, $\mathbf{W}_{\text{PP-SUB}}$, $\mathbf{V}_{\text{SC}}$, or $\mathbf{V}_{\text{CA1-SUB}}$) are composed of four submatrices, connecting the corresponding feature encoding sub-vectors (place-place, place-object, object-place, and object-object):

$$\mathbf{M} = \begin{bmatrix} \mathbf{M}^{\text{place,place}} & \mathbf{M}^{\text{object,place}} \\ \mathbf{M}^{\text{place,object}} & \mathbf{M}^{\text{object,object}} \end{bmatrix}. \tag{26}$$

Associations between objects and places were initially stored in $\mathbf{V}_{\text{SC}}$ as described below. To achieve a consistency in the code for places and objects, the weights in $\mathbf{V}_{\text{SC}}$ and $\mathbf{V}_{\text{CA1-SUB}}$ that connect neurons coding for the same feature (i.e., place-place or object-object) were set proportional to identity matrices $\mathbf{I}$,

$$\mathbf{V}_{\text{SC}}^{\text{place,place}} = \mathbf{V}_{\text{SC}}^{\text{object,object}} = w_{\text{SC}}^{\text{id}}\mathbf{I}, \tag{27}$$

$$\mathbf{V}_{\text{SC}}^{\text{place,place}} = \mathbf{V}_{\text{CA1-SUB}}^{\text{object,object}} = w_{\text{CA1-SUB}}^{\text{id}}\mathbf{I}. \tag{28}$$

The scaling factors $w_{\text{SC}}^{\text{id}} = \frac{1}{4}$ and $w_{\text{CA1-SUB}}^{\text{id}} = \frac{1}{2}$ ensure that these pathways had similar impact as the other pathways projecting to CA1 cells and SUB cells, respectively, and $w_{\text{CA1-SUB}}^{\text{id}}$ is twice as large as $w_{\text{SC}}^{\text{id}}$ to account for the fact that only in the CA1-SUB pathway the object-place and place-object connections were set to zero. The matrices $\mathbf{W}_{\text{PP-CA1}}$ and $\mathbf{W}_{\text{PP-SUB}}$, which represent shortcuts, were plastic during a consolidation cycle and evolved according to the learning rule described below. Their initial values were chosen as a random permutation of an equilibrium state, taken from a long running previous simulation.

**Place- and object-coding cells.** Place-coding cells in EC and DG-CA3 were assumed to respond deterministically, given a two-dimensional position variable $\mathbf{p}(t) \in [0, 1]^2$, which evolves in time.

Place-coding cells in entorhinal cortex show grid field spatial tuning [48], which we modelled as a superposition of 3 plane waves with relative angles of $\frac{\pi}{3}$:

$$\mathbf{x}_{\text{EC},i}^{\text{place}}(t) = r_{\max} \frac{2}{9} \sum_{l=1}^{3} \left[ \frac{1}{2} + \cos\left(m_i \mathbf{k}_i^l (\mathbf{p}(t) - \mathbf{p}_i)\right) \right], \tag{29}$$

where the spacing $m_i = 2\pi\left(2 + \frac{4i}{N}\right)$, $\forall i \in [1, N]$, is chosen so that a total range of 2 to 6 periods fit into the circular environment. The orientation of the plane waves is determined by the vector $\mathbf{k}_i^l = \left[\cos\left(l\frac{\pi}{3} + \theta_i\right), \sin\left(l\frac{\pi}{3} + \theta_i\right)\right]$ where $\theta_i$ are uniformly chosen random angles, and $\mathbf{p}_i \in [0, 1]^2$ are uniformly sampled random phases of the grid field [49]. Each cell's output rate varies between 0 to $r_{\max}$ spikes per second.

Place-coding cells in DG-CA3 show place-field tuning and were assumed to have a 2D Gaussian activity profile

$$\mathbf{x}_{\text{CA3},i}^{\text{place}}(t) = r_{\max} \exp\left( -\frac{(\mathbf{p}(t) - \mathbf{c}_i)^2}{2\sigma^2} \right), \tag{30}$$

where $r_{\max}$ is the maximum rate, $\sigma$ the field size, and $\mathbf{c}_i$ the centre of field $i$. The centres $\mathbf{c}_i$ were chosen to lie on a regular grid.

The object-coding cells in EC and DG-CA3 respond with fixed deterministic responses $\mathbf{x}_{\text{EC}}^{\text{object}}$ and $\mathbf{x}_{\text{CA3}}^{\text{object}}$ to each of $N_{\text{object}}$ objects. Given that they are located in the same brain region, we assumed that the firing-rate statistics of the object-coding cells and the place-coding cells were similar, both in EC and CA1. This was ensured by calculating the rates of the object-coding cells in two steps. First, we used the same equations as for the place-coding cells (i.e., Eq (29) for EC cells and Eq (30) for DG-CA3 cells) with a randomly selected "object position" $\mathbf{o}_i$, $i \in \{1, .., N_{\text{object}}\}$ for each of the $N_{\text{object}}$ objects. Subsequently the rates of the neurons within the population were randomly permuted for each object, to avoid an artificial constraint of the population activity onto a 2-dimensional manifold.

**Imprinting of place-object associations in the SC pathway.** The virtual animal learned a single new object-to-place association each day. Storing more memories per day would not qualitatively change the results, but would merely alter the time scale at which a given memory is overwritten in the SC pathway. Memories were imprinted in $\mathbf{V}_{\text{SC}}$ by first determining the activities of the object-coding DG-CA3 cells and place-coding CA1 cells given a random object and a random position where the object was encountered (see previous section). The weights in $\mathbf{V}_{\text{SC}}$ that connect object cells to place cells were then updated according to

$$\mathbf{V}_{\text{SC}} \leftarrow \left[ \mathbf{V}_{\text{SC}} + \frac{\lambda_{\text{SC}}[\mathbf{x}_{\text{CA3}}\mathbf{y}_{\text{CA1}}^{\mathsf{T}}]^{\text{norm}}}{1 - \lambda_{\text{SC}}} \right]^{\text{norm}}, \tag{31}$$

where $0 < \lambda_{\text{SC}} < 1$ (numerical values of parameters are summarized in Table 1) denotes the

strength of the new memory and controls the rate of forgetting. The symbol $[\mathbf{M}]^{\text{norm}}$ denotes the normalized version of the matrix $\mathbf{M}$; the normalisation ensures that the biggest sum along the columns of $[\mathbf{M}]^{\text{norm}}$ was 1 by rescaling all entries of $\mathbf{M}$ with the same factor. The specific choice of the normalisation does not alter the results. The inner norm in Eq (31) ensures the same relative influence of different memories, irrespective of the associated activity levels. This ensures an approximately constant rate of overwriting/forgetting. The outer norm guarantees that the weights $\mathbf{V}_{\text{SC}}$ stay bounded and hence induces forgetting. As a consequence of this updating scheme, the memories are lost over time. Note that before we imprint a new memory to $\mathbf{V}_{\text{SC}}$ (other than on day 0 on which the place-object association is learned that is tracked during the simulation), the place-coding cells in DG-CA3 are remapped, i.e., they are assigned to new random positions. This corresponds to learning the new object in a new environment/room, and effectively reduces the amount in interference between memories. Before starting a simulation, we imprinted $N_{\text{mem}}$ place-object associations to $\mathbf{V}_{\text{SC}}$ to ensure an equilibrium state.

The weights from place-to-object coding cells could be updated analogously. This would allow to decode the identity of a stored object given a location. We did not test this direction of the object-place association, because this is not relevant for the water maze task.

**Learning rule operating on $\mathbf{PP}_{\text{CA1}}$ and $\mathbf{PP}_{\text{SUB}}$ pathways.** The plastic weight matrices $\mathbf{W}_{\text{PP-CA1}}$ and $\mathbf{W}_{\text{PP-SUB}}$ changed according to a timing-based learning rule [41]:

$$\frac{\mathrm{d}\mathbf{W}}{\mathrm{d}t} = \int_0^\infty \mathrm{d}\tau [L(\tau)\, \mathbf{x}_{\text{EC}}(t-\tau)\, \mathbf{y}^{\mathsf{T}}(t) + L(-\tau)\, \mathbf{x}_{\text{EC}}(t)\, \mathbf{y}^{\mathsf{T}}(t-\tau)], \tag{32}$$

where $\mathbf{W}$ is either $\mathbf{W}_{\text{PP-CA1}}$ or $\mathbf{W}_{\text{PP-SUB}}$, and $\mathbf{y}$ correspondingly $\mathbf{y}_{\text{CA1}}$ or $\mathbf{y}_{\text{SUB}}$. The learning window $L(\tau)$ defined in Eq (12) determines the learning dynamics.

Eq (32) differs from the corresponding Eq (2) in several ways. First, on the left-hand side there is now a derivative, in contrast to the earlier version with a differential quotient; and on the right-hand side we omit the angular brackets that indicated a temporal average. Therefore, Eq (32) represents the instantaneous change of weights for a particular input, which is numerically more straightforward to implement in an online-learning paradigm. The resulting weight change for long times and many inputs approximates well Eq (2) if consolidation is slow enough. Second, we now omit the learning rate parameter $\eta$, which is absorbed in the definition of the parameters $A^+$ and $A^-$ of the learning window $L$. Third, there are now two addends in the integral and the integration limits are from 0 to $\infty$. This is equivalent to the earlier definition, but more convenient for a numerical implementation. All this allows to simplify the description of the learning dynamics, as will be outlined in what follows.

We integrated the learning dynamics using the Euler method, with time steps $\Delta T$ equal to the inverse pattern presentation rate. In practice, we used the standard method of calculating pre- and postsynaptic traces $\hat{\mathbf{x}}$ and $\hat{\mathbf{y}}$ to integrate the equation

$$\frac{\mathrm{d}\mathbf{W}}{\mathrm{d}t} = A^+\, \hat{\mathbf{x}}_{\text{EC}}(t)\, \mathbf{y}^{\mathsf{T}}(t) + A^-\, \mathbf{x}_{\text{EC}}(t)\, \hat{\mathbf{y}}^{\mathsf{T}}(t) \tag{33}$$

where $A^+$ and $A^-$ again determine the maximum amount of potentiation and depression of the synaptic weights, respectively. Note that these parameters effectively control the learning rate and are chosen twice as large in the $\text{PP}_{\text{CA1}}$ than in the $\text{PP}_{\text{SUB}}$ (Table 1), to increase memory lifetime in the latter shortcut. Again, we used an exponential window function $L(\tau)$, so that

exponentially filtered activities $\hat{\mathbf{x}}$ and $\hat{\mathbf{y}}$ can be calculated as in [98]:

$$\tau_{\text{STDP}} \frac{d\hat{\mathbf{x}}_{\text{EC}}(t)}{dt} = \mathbf{x}_{\text{EC}}(t) - \hat{\mathbf{x}}_{\text{EC}}(t) \quad \text{and} \quad \tau_{\text{STDP}} \frac{d\hat{\mathbf{y}}(t)}{dt} = \mathbf{y}(t) - \hat{\mathbf{y}}(t), \tag{34}$$

where $\tau_{\text{STDP}}$ determines the width of the learning window.

Weight values are constrained to the interval $[0, w_{\text{max}}]$. The weights of $\mathbf{W}_{\text{PP-CA1}}$ and $\mathbf{W}_{\text{PP-SUB}}$ were initialized to small random values from a uniform distribution in $[0, w_{\text{init}}^{\text{max}}]$.

For each iteration in a consolidation cycle of duration $T_c$, i.e., every $\Delta T = 5\text{ms}$, we chose a random input position and a random object to calculate the activities in all layers. These activities were then used to update the weights as given in Eq (33).

**Assessing the strength of memories in SC, PP$_{\text{CA1}}$, and PP$_{\text{SUB}}$.** To assess the memory strength encoded in a pathway, we determine the activity $\mathbf{y}^{\text{place}}$ of place-coding cells (in either CA1 or SUB) in response to an object $o \in \{1, \ldots, N_{\text{objects}}\}$ along the object-to-place pathway under consideration (e.g., for PP$_{\text{CA1}}$ it would be from object-coding cells in EC to place-coding cells in CA1). From this response we decode the memorized place of the object using Bayesian inference. However, the response is usually corrupted due to various factors such as imperfect imprinting, consolidation, or interference with other memories. Assuming that these imperfections result from a superposition of many statistically independent factors, we use a Gaussian likelihood:

$$p(\mathbf{y}^{\text{place}}|\mathbf{p}) = \mathcal{N}(\mu(\mathbf{p}), \sigma_{\text{noise}}\mathbf{I}), \tag{35}$$

where $\mathcal{N}$ is the multivariate Gaussian probability density function, $\sigma_{\text{noise}}$ is the standard deviation of the noise, i.e., the imperfections. $\mathbf{I}$ is the identity matrix, i.e., we assumed uncorrelated noise in the responses.

The expected activity $\mu(\mathbf{p})$ depends on the location $\mathbf{p}$ and is given by the activity that would result from the activation of place-coding cells in EC or DG-CA3, i.e., by Eqs (30), (23) and (24). Because the connections between place-coding cells in DG-CA3, CA1, and SUB are scaled identity matrices, the expected activity $\mu(\mathbf{p})$ is essentially a place-cell code:

$$\mu(\mathbf{p}) \propto \mathbf{x}_{\text{CA3}}^{\text{place}}(\mathbf{p}). \tag{36}$$

To avoid a dependence on overall activity levels, $\mu(\mathbf{p})$ and $\mathbf{y}^{\text{place}}$ are normalized to zero mean and unit variance.

Using Bayes' theorem we can now calculate the posterior probabilities of the places that coded for the given response $\mathbf{y}^{\text{place}}$:

$$p(\mathbf{p}|\mathbf{y}^{\text{place}}) = \frac{p(\mathbf{y}^{\text{place}}|\mathbf{p})p(\mathbf{p})}{\sum_{\mathbf{p}} p(\mathbf{y}^{\text{place}}|\mathbf{p})p(\mathbf{p})} \tag{37}$$

$$= \frac{p(\mathbf{y}^{\text{place}}|\mathbf{p})}{\sum_{\mathbf{p}} p(\mathbf{y}^{\text{place}}|\mathbf{p})} \quad \text{(assuming a flat prior)} \tag{38}$$

$$\propto \exp\left(\frac{-(\mathbf{y}^{\text{place}} - \mu(\mathbf{p}))^2}{2\sigma_{\text{noise}}^2}\right), \tag{39}$$

where for Eq (38) we used a flat prior, because the environment was uniformly sampled in the simulations. To avoid the explicit evaluation of the sum in the denominator, we normalise the evaluated place probabilities to sum to one. We make use of the linear relationship of the place

response given an object (see Eqs (23) to (26)):

$$\mathbf{y}^{\text{place}}(o) = \left(\mathbf{M}^{\text{object,place}}\right)^{\mathsf{T}}\mathbf{x}^{\text{object}}(o) \tag{40}$$

where the matrix $\mathbf{M}^{\text{object,place}}$ is either $\mathbf{V}_{\text{SC}}^{\text{object,place}}$, $\mathbf{W}_{\text{PP-CA1}}^{\text{object,place}}$, or $\mathbf{W}_{\text{PP-SUB}}^{\text{object,place}}$, depending on the pathway for which the strength of the memory is assessed. This allows to compute the posterior probability of the place given an object (Fig 4B and 4C):

$$p(\mathbf{p}|o) \propto \exp\left(\frac{-\left(\mathbf{y}^{\text{place}}(o) - \mu(\mathbf{p})\right)^2}{2N\sigma_{\text{noise}}^2}\right). \tag{41}$$

**Memory consolidation over many days.**   To simulate a single consolidation cycle (i.e., a storage of a new memory followed by a single consolidation phase), we alternated the imprinting of a new place-object association (Eq (31)) with a consolidation phase of length $T_c$. Before starting the experiments, we equilibrated the weights $\mathbf{W}_{\text{PP-CA1}}$ and $\mathbf{W}_{\text{PP-SUB}}$ by simulating $N_{\text{equi}}$ consolidation cycles. At day 0 we imprinted the object $\hat{o}$: the memory which was tracked. After each following consolidation phase the place probabilities along the different pathways were calculated for object $\hat{o}$ according to Eq (41) (see Fig 4C).

**Lesion experiments.**   Remondes and Schuman [27] lesioned the perforant path (temporoammonic pathway) during a Morris water maze consolidation experiment. Their finding evidenced a role of the perforant path in memory consolidation by showing that the precise timepoint of the lesion after memory acquisition determined whether the memory persisted (see Fig 4D).

In our simulations we implemented a lesion by setting all $\text{PP}_{\text{CA1}}$ weights to 0 ($\mathbf{W}_{\text{PP-Ca1}} = 0$) and by disabling their plasticity. Like in the experimental setup of [27], we lesioned either right before or 21 days after presentation of object $\hat{o}$. For each day and lesioning protocol, the place probabilities, Eq (41), along the pathways can then be calculated. The pathway with the highest inferred object position probability was then selected, and the summed probabilities per quadrant were calculated for this pathway. To account for exploration versus exploitation (see, e.g., [103]) of the rats, the inferred probabilities were linearly mixed with a uniform distribution over the quadrants. We used 70% explore versus 30% exploit for the plots in Fig 4D. Note that we assumed that the probabilities per quadrant correspond to the time spent in each quadrant.

## Consolidation in a hierarchical rate-based network

Fig 5 demonstrates the consolidation of memories in a hierarchy of connected neural populations. In the model, signals flow along distinct neocortical neural populations to the hippocampal formation (HPC) and back into neocortex (black arrows in Fig 5A). Shortcut connections exist between the neocortical populations (colored arrows in Fig 5A). All connections carry the same transmission delay $D$.

Every day new memories are imprinted into the weight matrix representing the HPC. The model describes the transfer of the memories into neocortex during $N_{\text{cycle}}$ consolidation phases, of which there is one per night (for all model parameters and values, see Table 2). In contrast to the model for Fig 4, we do not consider object-place associations, but directly analyse correlations between a stored memory weight matrix and the weight matrices that describe the neocortical shortcut connections.

**Model details.**   We consider a hierarchy of $2L$ neocortical populations with $L = 8$ shortcut connections. Activities of the populations that project towards the HPC are given by vectors $\mathbf{x}_i(t)$ and the activities of the populations leading away from the HPC by vectors $\mathbf{y}_i(t)$ ($i \in \{1, \ldots, L\}$). At each iteration, the activities $\mathbf{x}_L(t)$ (i.e., the neocortical population most distal

**Table 2. Parameters for simulations in Fig 5.**

| | | |
|---|---|---|
| $N_{\text{cycle}}$ | 1000 | number of consolidation cycles |
| $T_c$ | 150 s | consolidation time per sleep cycle |
| $\Delta T$ | 5 ms | integration time step |
| $N$ | 256 | neurons per neuron population |
| $L$ | 8 | number of neocortical populations |
| $r$ | 10 spikes/s | mean firing rate |
| $D$ | 5 ms | transmission delay |
| $\lambda$ | 0.5 | relative strength of new memory to HPC weights (see Eq 47) |
| $w_{\text{max}}$ | $2/N$ | maximum weight |
| $A_i^+$ | $0.4 \cdot w_{\text{max}} \cdot q^{i-1}$ | height of potentiating learning window for connections between populations at level $i$ |
| $A_i^-$ | $-1.00008 \cdot A_i^+$ | height of depressing learning window for connections between populations at level $i$ |
| $q$ | 0.5 | learning rate decrease factor |
| $\tau_{\text{STDP}}$ | 20 ms | time constants of learning window (see Eq 34) |
| $N_{\text{equi}}$ | 1000 | equilibration consolidation cycles run before the simulation starts |

from the HPC) are sampled from a Gaussian distribution with a mean input rate $r$ and a standard deviation $r/2$. The sampled activities are rectified to be non-negative ($r \leftarrow \max(r, 0)$), hence yielding a rectified Gaussian distribution. The activities on all other layers are then determined by their respective connections. For simplicity, we assume that weight matrices connecting subsequent populations in the hierarchy (black arrows in Fig 5A) are identity matrices that are scaled such that activity levels remain comparable along the hierarchy (see below). The results do not depend on this simplifying assumption. The population activities along the HPC directed path are then given as

$$\mathbf{x}_i = \mathbf{x}_{i+1}(t - D), \qquad \forall i \in \{1, .., L - 1\}. \tag{42}$$

In Fig 5, we modelled the HPC as a single neural population, with activities given by

$$\mathbf{y}_{\text{HPC}}(t) = \mathbf{V}_{\text{HPC}}^{\mathsf{T}} \mathbf{x}_1(t - D). \tag{43}$$

Here, $\mathbf{V}_{\text{HPC}}$ is the hippocampal-formation weight matrix into which new memories are imprinted (see below).

The first outward-directed neocortical population receives input from the HPC and through a shortcut connection from the activities $\mathbf{x}_1$,

$$\mathbf{y}_1(t) = \tfrac{1}{2}\mathbf{W}_1^{\mathsf{T}}\mathbf{x}_1(t - D) + \tfrac{1}{2}\mathbf{y}_{\text{HPC}}(t - D) \ . \tag{44}$$

Using Eq (43), we obtain

$$\mathbf{y}_1(t) = \tfrac{1}{2}\mathbf{W}_1^{\mathsf{T}}\mathbf{x}_1(t - D) + \tfrac{1}{2}\mathbf{V}_{\text{HPC}}^{\mathsf{T}}\mathbf{x}_1(t - 2D) \ . \tag{45}$$

Note that Eq (45) is slightly different from Eq (1) because we have included the delay $D$ now also in the direct pathway, for consistency; this does not influence the learning dynamics or the applicability of the theoretical analyses because the same delay is included in the learning rule in Eq (48). Subsequent activities $\mathbf{y}_i$ of populations projecting away from HPC are calculated as

$$\mathbf{y}_i(t) = \tfrac{1}{2}\mathbf{W}_i^{\mathsf{T}}\mathbf{x}_i(t - D) + \tfrac{1}{2}\mathbf{y}_{i-1}(t - D), \quad \forall i \in \{2, .., L\}, \tag{46}$$

where $\mathbf{W}_i$ are the direct shortcut connections from the populations $\mathbf{x}_i$ to the populations $\mathbf{y}_i$.

Memory imprinting to the HPC weight matrix $\mathbf{V}_{\text{HPC}}$ is analogous to the imprinting used in Fig 4 (compare Eq (31)). Before each consolidation phase, new memories were sampled from a binomial distribution $\mathbf{B}(1, 0.5)$. The HPC weights were then updated as

$$\mathbf{V}_{\text{HPC}} \leftarrow \left[ \mathbf{V}_{\text{HPC}} + \frac{\lambda [\mathbf{B}(1, 0.5)]_1^{\text{norm}}}{1 - \lambda} \right]_1^{\text{norm}}, \tag{47}$$

where $[\mathbf{M}]_1^{\text{norm}}$ denotes the L1 normalization of each row of the matrix $\mathbf{M}$ and $0 < \lambda < 1$ is the strength of a new memory.

All shortcut connections $\mathbf{W}_i$ showed plasticity similar to Eqs (33) and (34), i.e.

$$\frac{d\mathbf{W}_i}{dt} = A_i^+ \, \hat{\mathbf{x}}_i(t - D) \, \mathbf{y}_i^{\mathsf{T}}(t) + A_i^- \, \mathbf{x}_i(t - D) \, \hat{\mathbf{y}}_i^{\mathsf{T}}(t) \tag{48}$$

and

$$\tau_{\text{STDP}} \frac{d\hat{\mathbf{x}}_i(t)}{dt} = \mathbf{x}_i(t) - \hat{\mathbf{x}}_i(t) \quad \text{and} \quad \tau_{\text{STDP}} \frac{d\hat{\mathbf{y}}_i(t)}{dt} = \mathbf{y}_i(t) - \hat{\mathbf{y}}_i(t), \tag{49}$$

with parameters $\tau_{\text{STDP}}$, $A_i^+$, and $A_i^-$ specified in Table 2. Weights were constrained to the interval $[0, w_{\text{max}}]$ with $w_{\text{max}} = \frac{2}{N}$ and $N$ being the number of neurons per layer. Initial weights were drawn from a uniform distribution in this interval. To increase memory lifetime in the system, learning rates were decreased along the hierarchy such that the learning rate in layer $i$ is smaller than that in layer 1 by a factor $q^{i-1}$. Hence, layers closer to the HPC are more plastic than more remote layers.

Before starting the main simulation of $N_{\text{cycle}}$ consolidation cycles, we equilibrated the weight matrices by simulating $N_{\text{equi}}$ consolidation cycles.

**Assessing the strength of memories in neocortical weight matrices.** To assess the decay of memory in the system, a reference memory $\mathbf{V}_{\text{ref}}$, i.e. a specific realization from a row-normalized binomial distribution $\mathbf{B}(1, 0.5)$, was imprinted according to Eq (47) to $\mathbf{V}_{\text{HPC}}$ at time $t = 0$. The memory pathway correlation, i.e., the Pearson correlation of this reference memory with all shortcut weight matrices $\mathbf{W}_i$ was then calculated.

In analogy to the Methods on Fig 4, the maximum correlation (across layers) was taken as the overall memory signal of the system. This yields the power law in Fig 5B. The noise level indicated in Fig 5B is the standard deviation of the correlation between two random matrices drawn from a binomial distribution $\mathbf{B}(1, 0.5)$ and then row-normalized, both having sample size $N^2$. Considering the central limit theorem, the noise level will be approximately $1/N$.

## Theoretical analysis of hierarchical consolidation

As outlined in the Results and illustrated in Fig 5, the suggested consolidation mechanism can be hierarchically iterated and leads to power law forgetting when the learning rates in the various pathways are suitably chosen. To get a theoretical understanding of this behaviour, let us consider the architecture shown in the Fig 6A, which is a generalized version of Fig 5A. The network consists of a hierarchy of $N + 1$ input layers and $N + 1$ output layers. For mathematical simplicity, the network is assumed to be linear (in contrast to the model described in Fig 5A, which was nonlinear due to biologically motivated weight constraints), and the representation in the input layers is assumed to be the same, i.e., the weight matrices between the input layers (indicated in black in Fig 6A) are all simply the identity matrix (in contrast to the model described in Fig 5A where the identity matrices were also scaled). Similarly, we also assume that all weight matrices between the output layers are also the identity matrix. The mathematical derivations presented in the following can be generalized to arbitrary weight matrices both

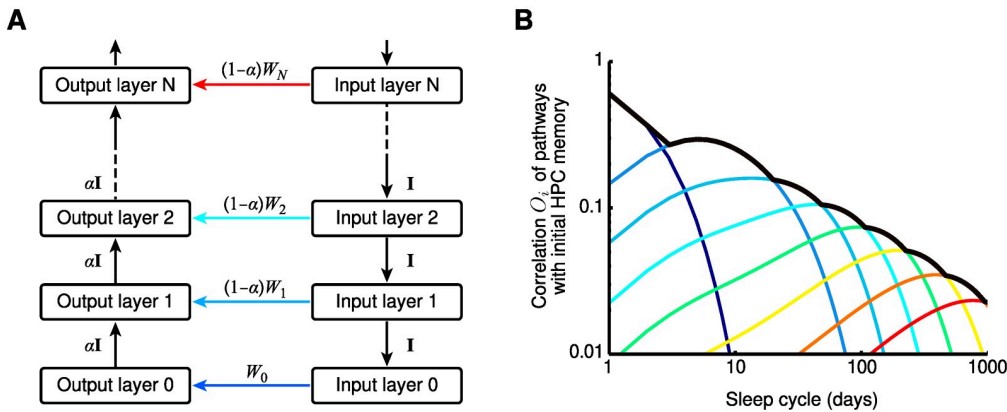

**Fig 6. Mathematical analysis of the hierarchical consolidation network.** (A) The mathematical analysis is performed for a network consisting of $N + 1$ input and $N + 1$ output layers. All output layers (except output layer 0) weight the input from the previous layer with a factor $\alpha$ and the input via the shortcut pathway with a factor $1 - \alpha$, to ensure that activity does not rise as increasingly many pathways converge onto the output layers. Input layer $i$ is hence connected to output layer $i$ through a shortcut connection with weight matrix $(1 - \alpha)W_i$ (except for the bottom-most layers $i = 0$, for which no factor $1 - \alpha$ is required). All connections between input layers are set to the identity matrix $\mathbf{I}$, and all connections between output layers are set to $\alpha\mathbf{I}$, for notational simplicity in the derivations. The math can be generalized to arbitrary connection matrices, as long as the network is linear. Each connection introduces a synaptic delay of $D$. The multi-synaptic pathway from input layer $i$ to output layer $i$ via shortcut connection $j \neq i$ has a total delay of $(2(i - j) + 1) \cdot D$, so the difference in delays between the pathway through shortcut $i$ and shortcut $j$ is $D_{ij} = 2(i - j) \cdot D$. (B) The similarity $O_i$ of the weight matrix $W_0$ (in which memory traces are initially stored) and the shortcut connection $W_i$ as a function of the time elapsed after storage (colored lines), and their maximum (black line). Simulations shown for $D = 2$ ms, $\alpha = 0.8$, $\eta_i = 2^{-i}$ and STDP time constant $\tau_{\text{STDP}} = 40$ ms.

in the input and the output pathways, but we prefer to treat the simple case to avoid cluttered equations and to make the theoretical approach more accessible.

We assume that due to newly acquired memories during the day, the weight matrix $W_0(t)$ (earlier called $\mathbf{V}_{\text{HPC}}$) that represents the memory trace in the hippocampus is varying in time, with an exponentially decaying autocorrelation function with time constant $\tau_{\text{overwrite}}$: $\langle \text{tr}(W_0(0)^{\mathsf{T}} W_0(t)) \rangle_t \propto \exp(-t/\tau_{\text{overwrite}})$, where tr denotes the trace of a square matrix.

All other pathways that project from an input layer to an output layer are plastic according to STDP. To derive the learning dynamics for these pathways, we first have to calculate the activity $\mathbf{y}_i$ in the $i$-th output layer,

$$\mathbf{y}_i(t) = \sum_{j=0}^{i} c_{ij} W_j^{\mathsf{T}} \mathbf{x}_j(t - D_{ij}) \,, \tag{50}$$

where $\mathbf{x}_j$ denotes the activity in input layer $j$ and $c_{ij}$ denote weighting factors that determine the impact of the $j$th pathway, i.e. the indirect pathway via $W_j$, on output layer $i$. These weighting factors are needed, because we would like to keep the weight matrices on a similar scale, but avoid that the activity increases from one output region to the next, because more synaptic pathways converge onto "later" output layers. The symbol $D_{ij} = 2D(i - j)$ (defined only for $i \geq j$) denotes the total additional delay that is accumulated on the connection from the $i$-th input layer to the $i$-th output layer that traverses the $j$-th direct "shortcut" pathway, relative to the direct shortcut from input layer $i$ to output layer $i$. For simplicity, we assumed that all connections have the same delay $D$. In a very similar way as in Eq (3), the learning dynamics of the

weight matrix $W_i$ in the direct path can be written as

$$\frac{\mathrm{d}W_i}{\mathrm{d}t} \approx \frac{\Delta W_i}{T} \approx \eta_i \sum_{j=0}^{i} c_{ij} \left[ \int L(\tau) \langle \mathbf{x}_i(t)\mathbf{x}_i^{\mathsf{T}}(t+\tau-D_{ij})\rangle_t \, \mathrm{d}\tau \right] W_j \tag{51}$$

where $\eta_i$ denotes the learning rate for the $i$-th pathway. For simplicity, we will assume that the different components of the input signal vector $\mathbf{x}_i(t)$ are uncorrelated amongst each other, and have identical temporal autocorrelations that are also independent of the layer index: $\langle \mathbf{x}_i(t)\mathbf{x}_i^{\mathsf{T}}(t+\tau)\rangle_t = \mathbf{I}f(\tau)$, where $\mathbf{I}$ is the identity matrix. The learning dynamics then simplify to

$$\frac{\mathrm{d}W_i}{\mathrm{d}t} \approx \eta_i \sum_{j=0}^{i} c_{ij} A(D_{ij}) W_j \tag{52}$$

with $A(D) := \int L(\tau)f(\tau - D) \, \mathrm{d}\tau$.

To measure the degree to which a memory trace that is stored in the weight matrix $W_0$ at time $t = 0$ is still present in the $j$-th shortcut pathway at a later time $t$, we compare the weight matrix $W_j(t)$ at time $t$ to the weight matrix $W_0(0)$ at time $t = 0$. We quantify the correlation of these two matrices by calculating the summed overlap of the column vectors:

$$O_i(t) = \mathrm{tr}[W_0(0)^{\mathsf{T}} W_i(t)] . \tag{53}$$

Note that the overlaps $O_i(t)$ are real numbers, and that their temporal dynamics for the shortcut connections (i.e., for all $i > 0$) are dictated by the dynamics of the weight matrices in the network:

$$\frac{\mathrm{d}}{\mathrm{d}t} O_i(t) = \mathrm{tr}\left[ W_0(0)^{\mathsf{T}} \frac{\mathrm{d}W_i(t)}{\mathrm{d}t} \right] \tag{54}$$

$$\approx \eta_i \sum_{j=0}^{i} c_{ij} A(D_{ij}) \, \mathrm{tr}[W_0(0)^{\mathsf{T}} W_j(t)] \tag{55}$$

$$= \eta_i \sum_{j=0}^{i} c_{ij} A(D_{ij}) O_j(t) . \tag{56}$$

To capture the exponential decay of the initially stored memories in the "hippocampal" weight matrix $W_0$ due to the storage of new memories, the set of dynamical equations is completed by

$$\frac{\mathrm{d}}{\mathrm{d}t} O_0(t) = -\frac{1}{\tau_{\mathrm{overwrite}}} O_0(t) . \tag{57}$$

Note that the dynamics of the overlaps $O_i$ form a linear dynamical system.

To show that this mathematical description shows a power-law behavior akin to the simulated system in Fig 5, we simulated the equations with the following parameter choices. Consistent with the exponential decay of the learning rates in the simulations, we chose the learning rates as $\eta_i = 2^{-i}$. The weighting factors $c_{ij}$ were chosen based on the assumption that output layer $i$ (for $i > 0$) receives a fraction $\alpha$ of its input from the output layer $i - 1$ below, and a fraction $1 - \alpha$ via its direct shortcut connection (associated with the weight matrix $W_i$). Taking into account that the signal reaching layer $i$ through shortcut connection $j$ traverses several of these weighting stages (Fig 6A), this choice yields $c_{ij} = \alpha^{i-j}$ for $j = 0$ and $c_{ij} = \alpha^{i-j}(1-\alpha)$ for

$j > 0$. Note that $\sum_{j=0}^{i} c_{ij} = 1$, so the activity level in different output layers should be similar. Finally, we assume that each synaptic transmission generates a fixed delay $D$ and that the auto-correlation function $f(\tau)$ decays much more quickly than the STDP learning window. In this case, we can approximate $A(D_{ij}) = \exp\left(-2D\frac{(i-j)}{\tau_{\text{STDP}}}\right)$.

For the simulations illustrated in Fig 6, we chose $\tau_{\text{STDP}} = 40$ ms as the time constant of an exponentially decaying STDP learning window for positive delays $\tau > 0$, and we set $A^+ = 1$ in Eq (12). Furthermore, we used $D = 2$ ms. As shown in the Fig 6B, the maximum of the overlaps $O_j$ indeed approximates a power law decay.

## Acknowledgments

We would like to thank Naomi Auer, Tiziano D'Albis, and Robert Gütig for discussions and feedback on the manuscript.

## Author Contributions

**Conceptualization:** Michiel W. H. Remme, Urs Bergmann, Susanne Schreiber, Henning Sprekeler, Richard Kempter.

**Data curation:** Denis Alevi.

**Formal analysis:** Michiel W. H. Remme, Urs Bergmann, Denis Alevi, Henning Sprekeler, Richard Kempter.

**Funding acquisition:** Michiel W. H. Remme, Susanne Schreiber, Henning Sprekeler, Richard Kempter.

**Investigation:** Michiel W. H. Remme, Urs Bergmann, Denis Alevi, Susanne Schreiber, Henning Sprekeler, Richard Kempter.

**Methodology:** Henning Sprekeler, Richard Kempter.

**Project administration:** Henning Sprekeler, Richard Kempter.

**Resources:** Susanne Schreiber, Henning Sprekeler, Richard Kempter.

**Software:** Michiel W. H. Remme, Urs Bergmann, Denis Alevi.

**Supervision:** Susanne Schreiber, Henning Sprekeler, Richard Kempter.

**Validation:** Denis Alevi.

**Visualization:** Michiel W. H. Remme, Urs Bergmann, Denis Alevi, Susanne Schreiber, Henning Sprekeler, Richard Kempter.

**Writing – original draft:** Michiel W. H. Remme, Urs Bergmann, Henning Sprekeler.

**Writing – review & editing:** Michiel W. H. Remme, Urs Bergmann, Denis Alevi, Susanne Schreiber, Henning Sprekeler, Richard Kempter.

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
