## [Decision Letter · Decision Letter 0]

7 Jun 2021

Dear Dr. Kempter,

Thank you very much for submitting your manuscript "Hebbian plasticity in parallel synaptic pathways: A circuit mechanism for systems memory consolidation" for consideration at PLOS Computational Biology.

As with all papers reviewed by the journal, your manuscript was reviewed by members of the editorial board and by several independent reviewers. In light of the reviews (below this email), we would like to invite the resubmission of a significantly-revised version that takes into account the reviewers' comments. In particular, the authors should ensure that the novel aspects of their model and predictions for future empirical studies are clearly and succinctly described in the main text.

We cannot make any decision about publication until we have seen the revised manuscript and your response to the reviewers' comments. Your revised manuscript is also likely to be sent to reviewers for further evaluation.

Sincerely,

Daniel Bush

Associate Editor

PLOS Computational Biology

Lyle Graham

Deputy Editor

PLOS Computational Biology

Reviewer's Responses to Questions

**Comments to the Authors:**

Reviewer #1: This manuscript presents a well-designed and neatly executed study of systems-level memory consolidation in spiking and rate neural networks. The authors identify a circuit motif and a simple learning rule that allows copying of information between the synaptic weights of an indirect and a direct pathway. The manuscript is well written, clearly organized, and describes the results in an easily understandable fashion.

I have some questions and concerns, listed below:

The authors briefly review the literature in the introduction, but in addition it would be useful to have a paragraph that clearly states what the authors perceive as the main novelty or scientific advance of the present work compared to other models of memory consolidation. Is it the fact the copying of information was achieved in a spiking model here? (This was also attempted in Tome et al: https://doi.org/10.1101/2020.12.22.424000). If so, it would seem appropriate to extend the study of spiking models beyond Figures 1 and 2. The discussion offers details on the relation of the present manuscript to previous phenomenological work, but not much comparison to prior work in the neural networks/computational neuroscience literature. More details on how the present copying mechanism is superior to or more biologically relevant than what has been done before would help.

One aspect in which the present work appears to differ is the focus on the particular circuit motif the authors discuss, which is suitable for feedforward, hetero-associative memory, while many other papers on systems-level consolidation focus on recurrent, auto-associative memory systems. This may make this study potentially more relevant to motor skill or habit acquisition than to consolidation of declarative memories - see below.

One major way in which the manuscript could be improved would be for the authors to show not merely that the copying of information is possible using their mechanism (which they do by monitoring the correlation of weights), but that this actually confers a computational advantage to a multi-stage memory system compared to a homogeneous memory system with the same number of synapses. In other words, when comparing memory systems of the same size (number of synapses), does breaking up the network into several subsystems and copying information between them actually increase the memory (recall) capacity? There are good theoretical arguments for why this should be the case (assuming the copying works well), and this should be easy to show at least for online learning with effectively binary weights (i.e., for strongly bimodal weight distributions), since it is known that such networks exhibit very poor (strongly sublinear) scaling of the memory capacity with system size. A more nontrivial (and interesting) test of the proposed mechanism would be to show that it can actually achieve a larger recall capacity than a homogeneous system of the same size that is well-equipped for good memory performance, namely one that uses sparse coding or complex synapses to achieve a favorable scaling of the memory capacity with system size.

(As an aside, my sense is that this manuscript really describes a mechanism rather than a conceptual advance, and personally I would hesitate to describe this as a “theory” in the scientific sense of the word. However, I understand that the word “theory” has been so frequently abused in the memory literature that the authors may feel that adding another “theory” to the list would do little harm.)

In the hippocampal example of Figs. 1 and 2, it is not entirely clear what computational goal consolidation serves, or how the proposed scheme fits in which the many ideas in the literature about the different functions of the SC and perforant pathways. The underlying assumption appears to be that place cells are learned from entorhinal grid cell input (it’s unclear if that’s in agreement with the experimental literature), and that this learned mapping from grid to place cells is then copied from the SC pathway to the perforant path. If this is the case, it would appear to make some strong predictions about place cells representations. In particular, when an animal returns to an environment it has experienced a long time ago, one may encounter a situation in which there would be no place cells in CA3 (the environment has been forgotten in the SC pathway), but there are place cells in CA1 (the environment is remembered in the perforant path). Has such a situation ever been observed experimentally?

Similarly in Fig. 3, it is encouraging that the authors can find model parameters that reproduce the behavioral effects of perforant path lesions. However, do we know experimentally whether in this (hippocampus-dependent) task the initial memory is in fact stored in the SC pathway, i.e., does an SC lesion shortly after task learning erase the memory trace? Conversely, does an SC lesion long after task learning leave the task performance unaffected, and preserve the CA1 place fields?

In Fig. 2F, is there a simple explanation for the dips in the PP_{CA1} tuning to the left and right of the peak?

As for the multi-stage consolidation simulations, it is of course well-known that essentially any process that generates an exponentially decaying memory signal on a certain (adjustable) timescale can be replicated with different timescales, and when these different timescale processes are read out with appropriate weights the corresponding exponential decays can be superimposed to approximate a power law. The same appears to apply to the system studied by authors in Fig. 4, and by choosing the parameters (learning rates and relative readout weights) appropriately, the system would presumably be able to generate overall memory signals that decay as power laws with various exponents, or as any number of other functions of time that don’t approximate power laws. Is there anything specific to the proposed system that would point to a particular preferred memory decay function, or allow a concrete prediction? Can anything universal be said about the effects of lesioning one particular set of plastic synapses, which led to forgetting in Fig. 3, but could perhaps also be compensated for if nearby pathways have similar forgetting timescales?

I understand that mechanistically the proposed multi-stage system requires appropriate time delays for the (very) indirect pathways in which the memory is initially stored, and one prediction is that the time delay for recall decreases during consolidation. This raises several questions: If the goal was in fact to eventually store information in a direct pathway with a short delay, why would the brain choose to initially store that information in an indirect pathway with a very long transmission delay? Is it reasonable to assume that achieving a short transmission delay is a central goal of memory consolidation? This may well be applicable to motor skill learning, or habit formation, but for the majority of the paper the authors frame their approach in terms of declarative memory. For episodic or semantic memory we don’t usually think of recall from a cue as immediately initiating an associated action, but more commonly recalled episodes and semantic information have to first be combined with other sensory inputs to inform decision making before any reasoned action can be taken. How can we reconcile this?

Finally, is the reduction in the delay of recall actually a necessary feature of systems-level consolidation? Clearly in the circuit motif the authors propose copying information using STDP works from the indirect to the direct pathway and not the other way around. However, it appears an “indirect” pathway in this study really just means a pathway with a longer transmission delay. As far as I can tell the authors imagine several layers of fixed weights interleaved with a single plastic layer in which memories are stored. The fixed weights are either not modeled at all, or simply modeled as identity transformations. Why would it have to be the case that the plastic layer in the indirect pathway is the last one, as in the SC pathway, or the middle one as in Fig. 4a? Could we not equally well make the first layer of the indirect pathway plastic and then introduce a delay by adding a layer of fixed weights afterwards? It would appear that copying information from such an indirect pathway to a direct one should work without problem at least if the fixed weights implement an identity transformation. However, in this situation the memory readout from the indirect and the direct pathways would actually have the same delay, since in both cases the memory resides in the first layer of weights after the input. If the speed of memory retrieval was a major concern, this would seem like a natural circuit to use.

Reviewer #2: The manuscript proposes a novel mechanistic theory of systems memory consolidation that spans multiple levels of detail, from single neurons to networks to multiple brain areas. The main idea of the theory is that original memories, stored in synaptic connections of one neural pathway (e.g. in the hippocampus), are progressively copied to shortcut neural paths up the neural hierarchy using standard hebbian synaptic plasticity rules. The functionality of the theory was demonstrated on the level of single spiking neurons, on the level of internal hippocampal networks modeled by populations of rate-based neurons and on the level of the whole brain in the context of classical phenomenological consolidation models. The proposed theory is shown to be consistent with experimental results in lesioned animals and with basic requirements of classical consolidation theories, and therefore it can provide a mechanistic basis for the earlier theories on the neuronal level providing a theoretical framework for testing and comparing them.

The strength of the proposed work is in the novelty, originality and depth of its primary proposal - the parallel pathway theory (PPT). Since the neuronal mechanisms underlying systems consolidation is a long-standing and important question in neuroscience, I believe that the proposed theory definitely deserves to be heard by a large audience of neuroscientists. The theoretical analysis is sound and rigorous.

However, a weak point of the paper, in my point of view, is that it is unnecessarily long and not optimally organised in terms of how the theory is demonstrated in simulations. While the main idea of the PPT is clear, it is presented using a number of different models (i.e. an integrate-and-fire model, a biophysical neuron model, and several rate-based models) and it is often not clear how all these models and their learning equations are related to each other. This diversity of models could have been justified by a unified underlying theoretical results, but the validity of analytical conclusions presented in the appendix for the simulations in the main text are not always clear. In other words, if the presented theory is primarily a computational theory, one would expect some theoretical statements to be valid for the whole range of models and simulations employed to illustrate how the theory works. In the manuscript, as it is written, it is not clear whether this is the case.

Another weak point is the lack of predictions from the proposed theory. How can this theory be experimentally verified? How is it possible to distinguish this theory from other competing theories? If this theory provides a mechanistic basis for classical phenomenological theories of systems consolidation, can it shed some light into the validity of generality of those classical theories? These issues should be addressed in the Discussion. More detailed comments are given below, more or less in the order of importance.

MAJOR ISSUES

1. The simulations (fig.1) and the associated theoretical analysis form the basis for the rest of the paper, but their presentation lacks clarity and seems underdeveloped. The theoretical analysis, as well as predictions derived from it, are for some reason put in the Appendix, whereas in my opinion it constitutes the core of the proposed hypothesis. Indeed, the authors claim to “propose a novel mechanistic foundation of the consolidation process” (line 43) and the link between the proposed consolidation mechanism and linear regression is an important theoretical insight in this respect, especially for this journal’s audience. Therefore, it seems reasonable to move the Supplementary material, or at least some part of it including the Eq. 43, into the main text with the required assumptions and the corresponding prediction (lines 646-648, see also point 4 below).

2. It would be helpful to more directly link the theoretical model and results in Supp. sections 1 and 2 to the simulation results in Fig.1. In particular, how does the spiking neuron model in Methods map to the theoretical model and in what conditions they are equivalent? What are the equivalents of W, V, x and x’, f and g of the theoretical model in the simulation? How do the simulation results in Fig.1 follow from the theoretical results (Eq. 43) ? Can they be directly compared in Fig.1?

3. The model of Section 3 of the results seems overly complicated in order to show an intuitively appealing result in Fig. 3C (bottom panel). Moreover, the relation of this rate model to the rate model theoretically analysed in earlier sections is not clear. What is the relation between the learning equation (Eq. 40) for the rate model in the Supp. Section 1 and Eq. 24 for the rate model in Methods? Logically it should be the same model, but it is not clear from the text if that’s the case. Do the theoretical results in Supp.Section 1 are applicable for the model of Fig.3 ?

It seems that it would be much easier to demonstrate consolidation/forgetting using the simulation setup in Fig.2 (i.e, memory of location in the linear track), but with the model of Fig.3A (or an equivalent rate model, e.g. that in the Supp. section 1). The introduction of object-place associations and two-dimensional grid-cells and place-cell codes seems unnecessary and makes the paper considerably longer, while it is not helpful to understand neither the Fig.3C (bottom panel) nor reproduce the experimental data in Fig. 3D.

4. Are there experimentally testable predictions from the proposed theory, e.g. on the level of hippocampal networks, apart from the prediction mentioned in the Appendix (lines 646-648) ?

Is there any evidence in the hippocampal replay literature supporting the prediction (lines 646-648) ? In one paper (Olafsdottir et al 2016, Nat Neurosci 19, 792–794 https://doi.org/10.1038/nn.4291), the replay in the EC was delayed with respect to CA1 by 11ms, which does not seem to be consistent with the model requirement.

It seems that the theory makes it possible to formulate necessary conditions on time delays and activity correlations between direct and indirect pathways that are necessary to implement consolidation (see e.g. caption of Supp.Fig2 B, i, ii and iii). It seems possible to experimentally verify all the three conditions for consolidation, unless the related data are already available in the literature. These issues can be addressed in the Discussion.

MINOR ISSUES

5. It is not clear whether the authors use the hippocampal formation just as a suitable example network to illustrate the functioning of their theory, or they argue for the specific role of the trisynaptic pathway in memory consolidation? Would it be possible to consolidate CA3 memories directly to EC-SUB connections, bypassing PP-CA1 ?

6. Direct EC-CA1 connections are sufficient to create CA1 place cells (Brun et al (2002). Place Cells and Place Recognition Maintained by Direct Entorhinal-Hippocampal Circuitry. Science, 296(5576), 2243–2246. https://doi.org/10.1126/science.1071089). Do these results invalidate the model ?

7. I do not see why the results section describes the hierarchical model in Fig 4, while the theoretical results are obtained for the model in Supp. Fig.3 ? Why not directly use the model in Supp.Fig 3?

8. Is there any evidence for “exponentially decreasing learning rates with distance from the hippocampus” (line 227), that seems to be a necessary condition for explaining the data by Wixted, 2004?

9. The authors mention several other models of synaptic memory consolidation (line 274). How does the proposed model compare to them? What are the distinguishing characteristics between the models? Is it possible to envisage experimental tests to see which model is more plausible?

10. Line 157, the reference to Morris and Lecar, 1981 is incorrect.

11. Typo, line 48 : thec -> the

Typo: In practise -> in practice (throughout the text)

**Have the authors made all data and (if applicable) computational code underlying the findings in their manuscript fully available?**

Reviewer #1: **No: **"The code will be made available upon publication."

Reviewer #2: **No: **The authors state that the code will be made available upon publication (page 45 of the manuscript), but it was not available at the moment of revision.

PLOS authors have the option to publish the peer review history of their article (what does this mean?). If published, this will include your full peer review and any attached files.

Reviewer #1: No

Reviewer #2: No
---

## [Decision Letter · Decision Letter 1]

24 Nov 2021

Dear Dr. Kempter,

We are pleased to inform you that your manuscript 'Hebbian plasticity in parallel synaptic pathways: A circuit mechanism for systems memory consolidation' has been provisionally accepted for publication in PLOS Computational Biology.

Best regards,

Daniel Bush

Associate Editor

PLOS Computational Biology

Lyle Graham

Deputy Editor

PLOS Computational Biology

Reviewer's Responses to Questions

**Comments to the Authors:**

Reviewer #1: Thank you very much for your replies to my comments.

Reviewer #2: The authors have thoroughly addressed all of the issues raised during the 1st revision.

**Have the authors made all data and (if applicable) computational code underlying the findings in their manuscript fully available?**

Reviewer #1: Yes

Reviewer #2: Yes

PLOS authors have the option to publish the peer review history of their article (what does this mean?). If published, this will include your full peer review and any attached files.

Reviewer #1: No

Reviewer #2: **Yes: **Denis Sheynikhovich

---

## [Editor Report · Acceptance letter]

1 Dec 2021

PCOMPBIOL-D-21-00486R1 

Hebbian plasticity in parallel synaptic pathways: A circuit mechanism for systems memory consolidation

Dear Dr Kempter,

I am pleased to inform you that your manuscript has been formally accepted for publication in PLOS Computational Biology. Your manuscript is now with our production department and you will be notified of the publication date in due course.

With kind regards,

Livia Horvath
